

# Measurement report: TURBAN observation campaign combining street-level low-cost air quality sensors and meteorological profile measurements in Prague

Petra Bauerová[1*], Josef Keder[1], Adriana Šindelářová[1], Ondřej Vlček[1], William Patiño[1], Jaroslav Resler[2], Pavel Krč[2], Jan Geletič[2], Hynek Řezníček[2], Martin Bureš[2,4], Kryštof Eben[2], Michal Belda[3], Jelena Radović[3,4], Vladimír Fuka[3], Radek Jareš[4], Igor Esau[5,6]

[1]Czech Hydrometeorological Institute, Na Šabatce 2050/17, 143 06 Prague 4, Czech Republic
[2]Institute of Computer Science, Czech Academy of Sciences, Prague, Pod Vodárenskou věží 271/2, 182 00 Prague 8, Czech Republic
[3]Charles University, Faculty of Mathematics and Physics, Ke Karlovu 3, 121 16 Praha 2, Prague, Czech Republic
[4]ATEM - Studio of ecological models, Roztylská 1860/1, 148 00 Prague 4, Czech Republic
[5]Nansen Environmental and Remote Sensing Centre, Jahnebakken 3, 5007 Bergen, Norway
[6]UiT - The Arctic University of Norway, Postboks 6050 Langnes, 9037 Tromsø, Norway

[*]*Correspondence to*: Petra Bauerová (petra.bauerova@chmi.cz)

**Abstract.** Within the TURBAN project, a "Legerova campaign" focusing on air quality and meteorology in the traffic-loaded part of the Prague city (Czech Republic) was carried out from 30 May 2022 to 28 March 2023. The network comprised of 20 combined low-cost sensor (LCS) stations for $NO_2$, $O_3$, $PM_{10}$ and $PM_{2.5}$ concentrations, along with a mobile meteorological mast, a single-channel microwave radiometer and Doppler LIDAR for measurement of vertical temperature and wind profiles. Significant individual deviations of LCSs were detected during the 165 day initial field test of all units at the urban background Prague 4-Libuš reference station (coefficient of variation 17-28 %). Implementing the Multivariate Adaptive Regression Splines method for correction reduced the LCS inter-individual variability and improved correlation with reference monitors in all pollutants ($R^2$ 0.88-0.97). The LCSs' data drifts and ageing were checked by the double mass curve method for the entire measurement period. During the Legerova campaign, the highest $NO_2$ concentrations were in traffic-loaded street canyons with continuous building blocks and several traffic lights. Aerosol pollution showed very little variation between the monitored streets. The highest $PM_{10}$ and $PM_{2.5}$ concentrations were recorded during temperature inversions and an episode involving pollution transported from a large forest fire in northern Czech Republic in July 2022. This report provides valuable data to support the validation of various predictive models dealing with complex urban environment, such as microscale LES model PALM tested in the TURBAN project.

**Graphical abstract**



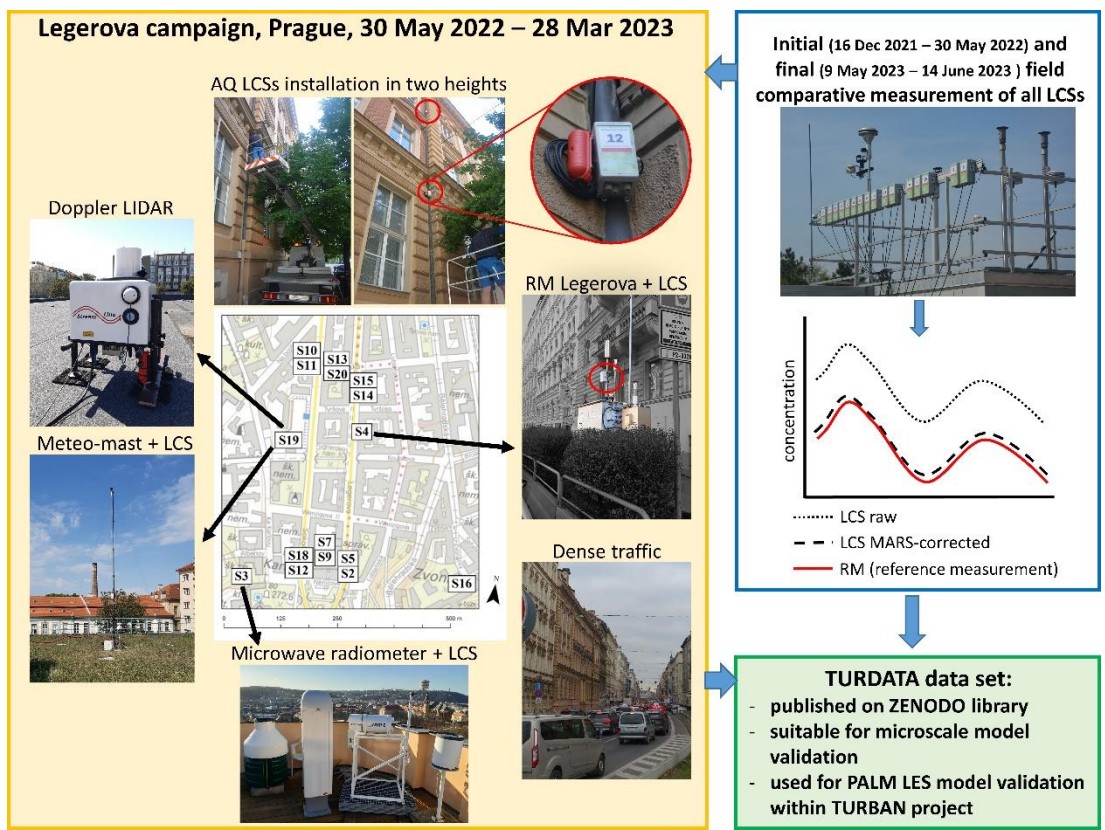

# 1 Introduction

With the growing importance of alerting the world to ongoing climate change and developing appropriate adaptation measures, the need to improve the modelling capabilities of meteorological and air quality conditions in the complex environment of large cities is increasing worldwide. To enable the improvement of atmospheric assessment and modelling in cities, it is always necessary to increase the spatial availability of measured or otherwise estimated data (i.e. remote sensing monitoring, satellite data). Since the reference meteorological and air quality monitoring (AQM) stations are technically demanding and often not

possible to relocate easily or relatively expensive to set up again for such targeted short-term observation campaigns, supplementary measurements in the form of readily available low-cost devices have become very popular in the last few years (Castell et al., 2017; Kumar et al., 2015; Morawska et al., 2018; Narayana et al., 2022).

Low-cost weather stations and air quality sensor stations are on the rise among the lay and interested public and even among scientists. These devices are therefore often used for citizen science and public data crowdsourcing (Jerrett et al., 2017;

Mahajan et al., 2020; Wesseling et al., 2019). The common problem is always the limited possibility of data quality control and assurance of these public data without previous or continuous control in comparison with national reference or equivalent measurements (RMs or EMs). In case of the low-cost sensors (further referred as LCSs) for ambient AQM, the data quality is





usually affected by the unstable performance of LCSs unit by unit within the same type and within a given LCS over time (Narayana et al., 2022; Peltier et al., 2020). One of the most commonly used is the electrochemical (EC) LCS for gaseous

pollutants, which is known for its limited lifetime (usually reported by manufacturers as between 12 and 15 months) due to the degradation of electrolyte performance over time and possible cross-sensitivities with some other gaseous substances (e.g. known interference between $NO_2$ and $O_3$; Baron and Saffell, 2017; Bauerová et al., 2020; Cui et al., 2021; Spinelle et al., 2015) and meteorological conditions, especially temperature and relative humidity (Bauerová et al., 2020; Collier-Oxandale et al., 2020; Jiao et al., 2016; Mead et al., 2013; Vajs et al., 2021). By contrast, the aerosol LCSs using optical particle counters

(OPCs) have a longer operational lifetime (usually between 2 to 3 years) and do not suffer from such a degree of unit-to-unit instability as the EC LCSs (Sayahi et al., 2019; Tagle et al., 2020). However, even the OPCs are known to interfere with meteorological conditions, especially with relative humidity and temperature (high probability of measurement error under condensation conditions). The mass concentration of the coarse fraction of aerosol particles ($PM_{10}$) is usually burdened by weaker performance in comparison with equivalent aerosol monitors (EMs) and by the greater probability of measurement

error with respect to relative humidity than the fine fraction $PM_{2.5}$ (Bauerová et al., 2020; Crilley et al., 2018; Tagle et al., 2020; Tryner et al., 2020). However, it is known that the error rate of the mass concentrations of all aerosol fractions depends mainly on the type of particle compounds and their properties with regard to the ability to bind water (Charron, 2004; Giordano et al., 2021; Robinson et al., 2023; Venkatraman Jagatha et al., 2021; Wang et al., 2021).

A general recommendation to overcome the given uncertainties and zero drifts is to undergo the following control process: 1)

a physical calibration of all the LCSs in a laboratory under controlled conditions; 2) verification according to the field comparative measurement of all LCSs with appropriate RMs or EMs at the AQM station (often called LCSs field calibration), followed by the application of a suitable statistical correction method (Clements et al., 2022; Peltier et al., 2020; Schneider et al., 2019; Spinelle et al., 2015). Nevertheless, to perform a calibration of each unit in a laboratory in the case of a large number of LCSs is relatively technically, financially and time-consuming for most end users (see for example Cui et al., 2021, or the

European standard CEN/TS 17660-1:2021 (E), 2021, for air quality of LCSs for gaseous pollutants). Moreover, it is known that only the laboratory calibration is not fully sufficient for successful LCS field deployment, because under controlled conditions it is not possible to demonstrate the changes of weather conditions and mixtures of gases and compounds occurring in the outdoor environment (De Vito et al., 2009; Kamionka et al., 2006). Therefore, some studies have already focused fully on sufficiently long field calibration for the evaluation of LCS performance (e.g. Cordero et al., 2018; deSouza et al., 2022;

Feinberg et al., 2018; Liu et al., 2020; Mukherjee et al., 2017). In this context, a very appropriate question is offered, namely what is sufficiently long field comparative measurement? The answer to this question is not clearly defined anywhere. Overall the recommendation based on the experience of different studies is, the longer the better. However, it is always necessary to balance the trade-off between the limited operating lifetime of the LCSs and the possibility to cover the widest possible range of meteorological and air quality conditions that may occur in the target deployment environment (Clements et al., 2022;

Giordano et al., 2021; Peltier et al., 2020). Therefore, the umbrella documents recommend a minimum of 30 to 40 days (CEN/TS 17660-1:2021 (E), 2021; Clements et al., 2022; Peltier et al., 2020; Yatkin et al., 2022a, b). Nevertheless, considering



the duration of one season in Central European conditions, this length does not ensure that a sufficiently wide range of meteorological conditions would be covered and therefore, it is even more desirable to repeat the control comparison tests after the season changes. Beside this, there are two main challenges in the case of long-term field LCS tests, namely random
data drifts and the possibility of performance changes after transfer to another location (De Vito et al., 2009; Papaconstantinou et al., 2023; Sayahi et al., 2019; van Zoest et al., 2019). The possible solution to treat this problem is performed either by a) repeated uninstallation of all LCSs and moving them to a single control RM site or b) providing a mobile reference unit and its temporary relocation for the mobile continuous field calibration of LCSs (Cui et al., 2021; De Vito et al., 2020; Venkatraman Jagatha et al., 2021). Unfortunately, both methods again increase the overall costs of low-cost measurement campaigns.
Therefore, the most common approach follows the general recommendation to collocate at least one sensor at the nearest RM station during the entire final deployment (CEN/TS 17660-1:2021 (E), 2021; Clements et al., 2022; Peltier et al., 2020; Yatkin et al., 2022a, b). The possible disadvantage of this procedure could be influenced by the individual performance of the given sensor (non-transferable individual error; De Vito et al., 2020; van Zoest et al., 2019).

To obtain the most reliable data from the LCSs measurements, it is always necessary to find an appropriate technique for
statistical correction of raw data. Due to the weaknesses of the LCSs as described above, it is evident that corrections based on single variable linear regression (i.e. only on the relationship between LCSs and RM- or EM-measured concentrations) may not be fully sufficient. Therefore, the multiple linear regression (MLR) analyses, generalised additive models (GAMs), random forests (RFs), K-nearest neighbours (KNNs), gradient boosting (GB), artificial neural networks (ANNs) and further complex algorithms, which allow taking into account other explanatory variables and non-linear relationships, are increasingly used,
achieving different levels of final LCS performance (Considine et al., 2021; deSouza et al., 2022; Kumar and Sahu, 2021; Mahajan et al., 2020; Narayana et al., 2022). In any case, it is highly desirable that the applied correction method is sufficiently transparent and computationally reproducible (avoiding black box methods), which is not always possible in the case of some new statistical machine learning techniques (e.g. random forests, neural networks). From this point of view, the multivariate adaptive regression splines (MARS) method can be a suitable statistical tool for LCS measurement correction since it is a non-
parametric regression technique that can reflect non-linearities and different interactions between continuous or categorical data (Friedman, 1991). MARS models are very flexible and simple to understand and interpret. Moreover, this method often requires little or no data preparation (capable of dealing with noisy data) and is computationally time-feasible and reproducible (Friedman, 1991; García Nieto and Álvarez Antón, 2014; Keshtegar et al., 2018). A common challenge when using different machine learning techniques is the possible loss of accuracy due to the incompleteness of the initially defined model, leading
to 'concept drift' (De Vito et al., 2020; Ditzler et al., 2015). Therefore, it is still recommended to perform continuous or backward controls of the performance of any correction algorithm used (similar to the previously mentioned LCS data drift control). Several data control mechanisms have already been described in papers focusing on LCS measurement (De Vito et al., 2020; Harkat et al., 2018). However, to our knowledge, no previous study has used the double mass curve (DMC) method. The DMC is a simple graphic method usually used for checking the consistency of hydrological and climatological data



continuously measured at several stations in a selected area (Kliment et al., 2011; Liu et al., 2023). According to our experience, we assume that it is fully applicable even to the control of the performance of the LCS network measurement.

For the possibility of a better understanding of complex atmospheric processes in the urban environment (including the accumulation and dispersion of pollutants), it is important to obtain data from different heights, not only within the urban canopy layer but also above it. Therefore the combination of the traditional ground measurement with remote sensing

monitoring of temperature and wind profiles above the rooftops is beneficial (Allwine et al., 2002; de Arruda Moreira et al., 2020, 2018; Münkel et al., 2007). The advantage of using microwave radiometers (MWR for temperature profiles) and doppler light detection and ranging systems (LIDARs for wind profiles) nowadays is their high temporal resolution (compared to radiosondes), portability and the possibility of installation in the city without disturbing the surroundings (in contrast to acoustic wind profilers or SODAR-RASS systems; Lokoshchenko et al., 2009; Tamura et al., 2001). However, even these

devices are burdened by their technical limitations and some data verification is recommended (if not against the available RM, at least compared to other remote sensing measurements). The Doppler LIDARs' accuracy of wind measurement can be deteriorated by rain (and low stratus clouds) and profiles have high vertical resolution but non-stable height range because of the varying signal-to-noise ratio (SNR). The MWRs measurement performance is quite independent of meteorological conditions (with some exceptions in older instruments as in Ezau et al. (2013), moreover, MWRs have null overlap and do not

use aerosols as tracers. On the other hand, MWRs usually have a stable height range, but a lower vertical resolution than LIDARs (de Arruda Moreira et al., 2020, 2018).

The objective of this study was to obtain credible air quality and meteorological data using a high spatio-temporal resolution supplementary network consisting of air quality LCSs, MWR and Doppler LIDAR in a selected urban environment to support the validation of the updated LES PALM microscale model. For the implementation of the TURBAN street-level observation

campaign and subsequent microscale modelling, the part of the Prague city centre within Legerova, Sokolská and Rumunská streets and their surroundings (Prague 2 district, the Czech Republic) was chosen. This area is a typical urban environment (within Central European cities) with high traffic loaded street canyons (the traffic intensity is between 35 and 45 thousand cars per day; TSK, 2023). The Prague 2-Legerova reference AQM station is classified as a hotspot (see NO, $NO_2$ and $NO_x$ measurement statistics across all traffic stations in Prague in Table S1 in the Supplement). Therefore, it is also one of the most

frequent target locations of public attention and protest actions for limiting automotive traffic (and automotive speed) in Prague. One of the main goals of the TURBAN project is to develop a new modelling tool validated against in situ and remote sensing measurement for supporting the Prague municipality and other interested entities in urban planning. Further, the project aims to prepare adaptation and mitigation strategies addressing mainly those issues associated with urban heat waves and degraded air quality (following the experience gained in previous studies by Resler et al., 2017, 2021). The most challenging

issue was resolving the quality of the LCSs' data. Some innovative procedures based on long-term field testing were applied: MARS statistical method for correction and identification of possible data drifts by DMC. All methods used are fully described in this paper and are reproducible. The TURDATA data set (a database of low-cost air quality and remote sensing measurements for the validation of micro-scale models in the real Prague urban environments) is publicly available at the



Zenodo library (Bauerová et al., 2024). The manuscript is structured according to the five main objectives of this paper: 1)
Evaluate the performance of raw LCS measurement and calculate appropriate MARS corrections for all LCSs; 2) Apply the
MARS correction equations on the raw LCS measurement during the Legerova measurement campaign and monitor the
measured and corrected data quality; 3) Evaluate the performance of supplementary meteorological measurements; 4) Analyse
and describe the air quality measurement during the Prague Legerova campaign; 5) Show interesting episodes during the
campaign from the point of view of air quality and meteorological profile measurement.

## 2 Materials and methods

### 2.1 Study area and experimental design

The measurement campaign for the TURBAN project in Prague comprised several stages taking place at two different localities
in Prague. The target deployment site for the observation campaign was named the Prague Legerova domain here (Fig. S1 in
the Supplement) and took place in the vicinity of the urban traffic hotspot Prague 2-Legerova air quality monitoring (AQM)
station (locality code: ALEGA; CHMI, 2023a) and the adjacent Prague Karlov professional meteorological station (MS) placed
on the roof of the building (station ID: P1PKAR01; WMO, 2023a). Within the Prague Legerova campaign 20 combined LCSs
(for $NO_2$, $O_3$, $PM_{10}$ and $PM_{2.5}$ concentrations) were installed in selected streets of Sokolská, Legerova and Rumunská and their
immediate surroundings (district of Prague 2, Czech Republic; Fig. 1b and Fig. 2). Of these, 11 LCSs were placed in the streets
with the highest traffic load: 10 LCSs were installed in pairs at two different height levels in five locations and one LCS
(identified as S4) was collocated with the Prague 2-Legerova traffic RM station throughout the entire campaign. Furthermore,
5 LCSs were installed at greater distances and higher heights from these streets and were established as background locations:
two LCSs on the roofs of the Prague Karlov MS (S3) and Le Palais Art Hotel Prague (S16), two LCSs (S7 and S9) within the
closed school courtyard (a student sports field with no traffic) and one LCS (S19) at about 50 m away from the middle of
Sokolská street with a high traffic load. Most of the LCSs started to measure in target locations from 30 May 2022 (except
LCSs S3, S4 and S16). Furthermore, the measurement campaign consisted of one mobile meteorological mast (MM; for basic
meteorological measurement below the level of the rooftop) installed in the garden of the Prague Waterworks and Sewerage
Company (hereinafter referred as the PVK garden), one MWR (for temperature vertical profile) installed on the roof of the
Prague Karlov MS and one Doppler LIDAR (for wind vertical profile) installed on the roof of the PVK administrative building
(hereinafter referred to as the PVK roof). Complete list of instruments, their placement details, measurement start dates and
other metadata are given in Table 1, locations are shown in Fig. 1b.

The second site was the Prague 4-Libuš suburban background AQM station (locality code: ALIBA; CHMI, 2023b) with the
adjacent Prague Libuš professional MS (station ID: P1PLIB01; WMO, 2023b), both located outside of the Legerova target
domain. At this locality, the initial and final field comparative measurement of all 20 combined LCSs took place before and
after the end of the Legerova measurement campaign. The initial field comparative measurement lasted for most of the LCS
stations (17 out of 20) from 16 December 2021 until 30 May 2022 (165 days) and the goal of this measurement was to identify





errors and deviations in measurements between individual LCSs and between LCSs and the reference or equivalent monitors (RM or EM; see Fig. 3). Finally, based on this dataset, the correction equations using a multivariate adaptive regression spline (MARS) method were calculated and applied to the raw measured LCS data. In 2 exceptions, the initial field comparative measurement lasted for a reduced period, namely until 23 February 2022 in the case of LCS S3 (69 days) and until 24 March

2022 for LCS S4 (98 days) given the earlier installation of these sensors in the Legerova domain, namely at the Prague Karlov MS roof (LCS S3) and at the Prague 2-Legerova AQM station (S4). Two sensors (S8 and S17) identified as faulty during the initial field comparative measurement were later replaced, compared on separate dates and left together with LCS S6 at the Prague 4-Libuš AQM station throughout the entire campaign. These units later served as verified spares in case of failures of other sensors during the main campaign.


**Table 1.** List of all measuring instruments used in the TURBAN project with the metadata to specific deployment locations. Coordinates are given in decimal degrees. Station classification follows the classification method used in air quality RM network (CHMI, Czech Republic).

| Station name | Station ID | Final deployment location name | Latitude | Longitude | Ground elevation (m ASL) | Height (m AGL) | Station classification[a] | Measurement start date |
|---|---|---|---|---|---|---|---|---|
| AQ Sensor 2 | S2 | school Legerova (lower height) | 50.069833 | 14.43075 | 237 | 5.8 | T | 30/05/2022 |
| AQ Sensor 5 | S5 | school Legerova (higher height) | 50.069833 | 14.43075 | 237 | 13.2 | T | 30/05/2022 |
| AQ Sensor 9 | S9 | School courtyard (lower height) | 50.069778 | 14.430389 | 237 | 4.7 | UB | 31/05/2022 |
| AQ Sensor 7 | S7 | School courtyard (higher height) | 50.069778 | 14.430389 | 237 | 6.9 | UB | 31/05/2022 |
| AQ Sensor 12 | S12 | school Sokolská (lower height) | 50.06975 | 14.429694 | 236 | 5.9 | T | 30/05/2022 |
| AQ Sensor 18 | S18 | school Sokolská (higher height) | 50.06975 | 14.429694 | 236 | 13.1 | T | 30/05/2022 |
| AQ Sensor 8 | S8 | RM Libuš | 50.0073 | 14.44593 | 301 | 2.5 | SUB | 22/05/2022 |
| | | school Sokolská (higher height) | 50.06975 | 14.429694 | 236 | 13.1 | UB | 15/02/2023 |
| AQ Sensor 14 | S14 | Legerova (lower height) | 50.073472 | 14.430278 | 236 | 9.2 | T | 30/05/2022 |
| AQ Sensor 15 | S15 | Legerova (higher height) | 50.073472 | 14.430278 | 236 | 14.6 | T | 30/05/2022 |
| AQ Sensor 20 | S20 | Rumunská (lower height) | 50.073611 | 14.430028 | 236 | 4.6 | T | 30/05/2022 |





| AQ Sensor 13 | S13 | Rumunská (higher height) | 50.073611 | 14.430028 | 236 | 14.8 | T | 30/05/2022 |
|---|---|---|---|---|---|---|---|---|
| AQ Sensor 11 | S11 | CKAIT Sokolská (lower height) | 50.073722 | 14.429139 | 235 | 5.5 | T | 30/05/2022 |
| AQ Sensor 10 | S10 | CKAIT Sokolská (higher height) | 50.073722 | 14.429139 | 235 | 12.2 | T | 31/05/2022 |
| AQ Sensor 19 | S19 | PVK garden (on meteo-mast) | 50.072111 | 14.428806 | 241 | 2.6 | U | 01/06/2022 |
| AQ Sensor 3 | S3 | MS Karlov (roof) | 50.069157 | 14.427839 | 235 | 30 | UB | 23/02/2022 |
| AQ Sensor 16 | S16 | Hotel Le Palais Art Prague (roof) | 50.069854 | 14.434532 | 238 | 22 | UB | 19/07/2022 |
| AQ Sensor 4 | S4 | RM Legerova | 50.072361 | 14.430667 | 238 | 2.1 | T (hotspot) | 24/03/2022 |
| AQ Sensor 6 | S6 | RM Libuš | 50.007305 | 14.445933 | 301 | 2.5 | SUB | 16/12/2021 |
| AQ Sensor 17 | S17 | RM Libuš | 50.007305 | 14.445933 | 301 | 2.5 | SUB | 16/12/2021 |
| Doppler LIDAR | LDR | PVK roof | 50.072588 | 14.428428 | 235 | 4.5 | U | 24/03/2022 |
| Microwave radiometer | MWR | MS Karlov roof[b] | 50.069157 | 14.427839 | 235 | 29.5 | UB | 23/02/2022 |
| Meteo-mast PVK | MM | PVK garden | 50.072111 | 14.428806 | 241 | 7.5[c] | U | 01/06/2022 |
| AQM Prague 2-Legerova | ALEGA | RM Legerova | 50.072388 | 14.430673 | 238 | 3.5 | T (hotspot) | continuous |
| AQM Prague 4-Libuš | ALIBA | RM Libuš | 50.007304 | 14.445933 | 301 | 3.5 | SUB | continuous |
| AQM Prague 9-Vysočany | AVYNA | RM Vysočany | 50.1110803 | 14.5030956 | 207 | 3.5 | T | continuous |
| MS Prague-Karlov | P1PKAR01 | MS Karlov roof[b] | 50.069167 | 14.427778 | 235 | 28.5 | UB | continuous |
| MS Prague-Libuš | P1PLIB01 | MS Libuš | 50.007778 | 14.446944 | 302 | 10 | SUB | continuous |

[a]T = urban traffic, UB = urban background, SUB = suburban background, U = urban; [b]MS Karlov is placed on the top of the building roof at 29.5 m AGL; [c]The height of wind measurement at the top of the meteorological mast.



**Figure 1.** (a) Map of the Czech Republic with the city of Prague and the locality of Hřensko highlighted; (b) Map of Prague city with selected Legerova location (highlighted in red rectangle), reference monitoring station and meteorological station Libuš and reference monitoring station Vysočany; (c) Map of the individual device placement within the TURBAN measurement campaign in Legerova street and its surroundings. Sx = individual low-cost sensors for air quality monitoring (AQ LCS), L = lower height ALG, H = higher height ALG. State boudaries in (a) © EuroGeographics. Background data in (a) and (b) is an Open Street Map provided through WMS by terrestris GmbH & Co. KG. Ortofoto map in (c) is provided through WMS by the Czech Office for Surveying, Mapping and Cadastre – ČÚZK.



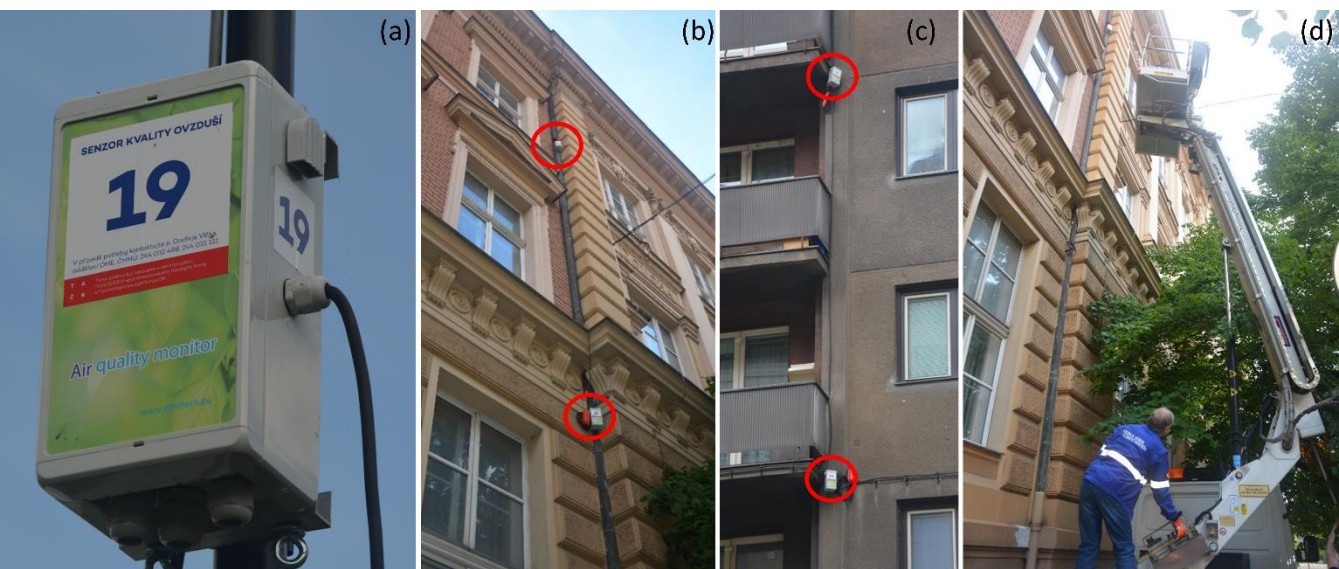

**Figure 2.** Photos of LCSs deployed in the Legerova observation campaign (Prague, Czech Republic). (a) The detailed picture of LCS stations used in TURBAN project for monitoring $NO_2$, $O_3$, $PM_{10}$ and $PM_{2.5}$ concentrations; (b) and (c) installation of LCSs at two different height levels at Sokolská school location and Legerova location; (d) picture of the lift platform used for installation of LCSs at different heights.

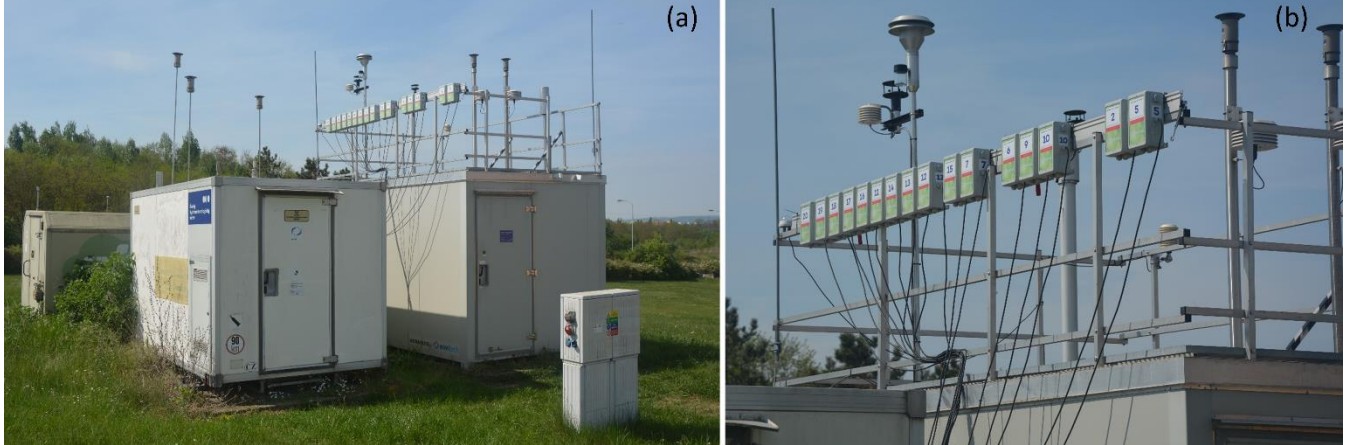

**Figure. 3.** (a) Initial field comparative measurement of all LCSs (for measuring $NO_2$, $O_3$, $PM_{10}$ and $PM_{2.5}$) at the Prague 4-Libuš AQM station (CHMI, 2023b); (b) LCSs in detail. Measurement lasted from 21 December 2021 to 30 May 2022.


## 2.2 Technical specification of all instruments used and measurement methodology

### 2.2.1 Air quality monitoring instruments

For the air quality monitoring, 20 combined LCS enviSENS platforms (Envitech Bohemia, CZ) were used. They were constructed as small airflow boxes with dimensions of 125 × 225 × 110 mm, each equipped with an $NO_2$ and combined $O_3/NO_2$



Cairsens electrochemical LCS (FR; Envea, 2023) and low-cost aerosol particle counter PMS7003 (CN; Plantower, 2023) for the measurement of $PM_{10}$ and $PM_{2.5}$ mass concentrations (for the main technical parameters see Table 2). These types of LCSs were chosen on the basis of our previous experience with an almost one-year testing field comparative measurement (Bauerová et al., 2020). All LCS stations were powered by 230 V electricity and the data were transferred remotely via LTE modems to the internal server of CHMI. The measurement frequency was set to 10-minute intervals in all sensors, from which 1-hour

averages were calculated. Furthermore, the data from RMs and EMs measuring at AQM stations Prague 2-Legerova and Prague 4-Libuš were used as a control. These stations are equipped with the RMs Teledyne API T200 for $NO_2$ monitoring and T400 for $O_3$ monitoring (US; Teledyne API, 2023a, 2023b) and with the EMs Palas Fidas 200 S (DE; Palas, 2023) optical particle counters for the measurement of $PM_{10}$ and $PM_{2.5}$. Since RM for $O_3$ measurement is not available at the Prague 2-Legerova urban traffic station, as a substitute $O_3$ concentrations measured at the Prague 9-Vysočany AQM station (also

classified as an urban traffic station; CHMI, 2024c) were used for indicative comparison with $O_3$ LCS measurement during the Legerova campaign. The technical parameters of all used devices are listed in Table 2.

**Table 2.** Technical parameters of LCSs and reference and equivalent monitoring (RM and EM) methods used for air quality measurement within the TURBAN project. Source of parameters Envea, 2023; Palas, 2023; Plantower, 2023; Teledyne API, 2023a, 2023b.

| Instrument type | Measured pollutant | Measurement principle | Measurement range | Limit of detection | resolution | Uncertainty | Interference effect | Temperature effect on zero value |
|---|---|---|---|---|---|---|---|---|
| **Cairsens Envea NO₂ (LCS)** | $NO_2$ | electrochemical gas sensor | 0-250 ppb | 20 ppb | 1 ppb | <30 % | $Cl_2$: ~ 80 % sulphur compounds: negative interference | ±50 ppb |
| **Cairsens Envea O₃/NO₂ (LCS)** | $O_3$ ($O_3$+oxidant) | electrochemical gas sensor | 0-250 ppb | 20 ppb | 1 ppb | <30 % | $O_3$: ~ 80 % $Cl_2$: ~ 80 % sulphur compounds: negative interference | ±50 ppb |
| **Plantower PMS7003 (LCS)** | $PM_1$ | optical particle counter | 0-1,000 µg·m⁻³ | 0.30 µm | 1 µg·m⁻³ | ±10 % at conc. 100-500 µg·m⁻³±10 µg·m⁻³ at conc. 0-100 µg·m⁻³ | Temperature and relative humidity | |
| | $PM_{2.5}$ | | 0-1,000 µg·m⁻³ | | 1 µg·m⁻³ | | | |
| | $PM_{10}$ | | 0-1,000 µg·m⁻³ | | 1 µg·m⁻³ | | | |
| **Teledyne API T200 (RM)** | $NO/NO_2/NO_x$ | chemiluminescence analyser | 0-20,000 ppb | <0.2 ppb | | 0.5 % of reading above 50 ppb | | |
| **Teledyne API T400 (RM)** | $O_3$ | UV absorption analyser | 0-10,000 ppb | <0.4 ppb | | 0.5 % of reading above 100 ppb | | |
| **Palas FIDAS 200S (EM)** | $PM_1$ | optical particle counter | 0-10,000 µg·m⁻³ | 0.18 µm | 0.1 µg·m⁻³ | | | |
| | $PM_{2.5}$ | | | | | 9.7 % for $PM_{2.5}$ | | |
| | $PM_4$ | | | | | | | |
| | $PM_{10}$ | | | | | 7.5 % for $PM_{10}$ | | |




### 2.2.2 Meteorological measurement instruments

As a supplementary non-reference measurement, the EnviMET mobile telescopic MM (Envitech Bohemia, CZ) was installed on 1 June 2022 in the PVK garden (see Fig. 4a). This MM was equipped with a 2D ultrasonic anemometer WindSonic 60 (Gill Instruments, UK) for wind velocity (WV) and wind direction (WD; for technical details see Gill, 2023a) placed at a height of

7.5 m above the ground and further with the MetConnect THP weather station (Gill Instruments, UK) for temperature (TMP), relative humidity (RH) and atmospheric pressure (p) placed at 2 m above the ground (for technical details see Gill 2023b). The measurement frequency was set to 10-minute intervals in all variables and later averaged to 1-hour data.

The vertical profiles of TMP were measured with the MTP-5-He MWR (Attex, RU) installed on the roof of the Prague Karlov MS (Fig. 4b) on 23 February 2022. The MTP-5-He is a single channel passive MWR measuring at a frequency of 56.6 GHz

with a maximum height range of 1,000 m and height resolution of 25 m from 0 to 100 m and 50 m from 100 to 1,000 m. The TMP accuracy is ±0.3 °C to ±1.2 °C (IFU, 2023). The measurement frequency was set to one vertical profile every 5 minutes. Beyond the measured TMPs, the mean TMP gradient and potential TMP profiles were calculated for further data processing. The TMP gradients were calculated from the mean difference in 1-hour average TMPs measured between ground level and 200 m height above the ground. The profiles of the potential TMP were calculated based on Arya (2001; see method description

in Sect. S2.1 in the Supplement) in the height layer between 260 and 1,260 m ASL with the reference pressure at the Prague Karlov MS (altitude 260 m ASL).

Furthermore, the radial velocity and backscatter intensity were measured by a Doppler LIDAR StreamLine XR (HALO Photonics, UK) installed on the PVK roof from 24 March 2022 (Fig. 4c). This Doppler LIDAR has an all-sky scanning head (full hemispherical coverage with 0.01° resolution in both axes) with the possibility of a variable user setting of laser pulse

directions. The maximum height range is 1,200 m (highly dependent on the specific scan mode setting). The pulse rate is 10 kHz and the velocity precision <0.2 m·s$^{-1}$ for signal-to-noise ratio (SNR) > -17 dB (Metek, 2023). The following three scanning modes were set: i) the VAD 6 mode with an elevation angle of 75° and azimuth step 60° for gaining processed vertical profiles of WV and WD; ii) the custom mode called "user 1" scanning a sector between azimuths of 130° to 160° while the elevation angle was gradually adjusted to values of 35° to 50° with a step of 5°; iii) the custom mode called "TKE" scanning a cone with

an apex angle of 109.48° (the recommended elevation angle is 35.26°) in the continuous scanning mode (CSM; according to the method Smalikho and Banakh (2017) and the angular velocity was set to 5 deg·s$^{-1}$ (i.e. one rotation of 360° takes approximately 72 seconds). The total probing cycle for all scanning modes was set to 30 minutes, so for the TKE mode itself approximately 25 rotations were made in the meantime. The data gained were further processed before usage (see Sect. 2.3.3).

In addition to the measurement listed above, TMP, RH, WD, WV and global radiation intensity (GLRD) data from MS Prague

Libuš and Prague Karlov were used (especially for the correction of LCS data). The Karlov MS is also equipped with a CL51 Vaisala ceilometer (FI; Vaisala, 2022) measuring continuously the cloud base heights, backscatter intensity profile and mean mixing layer height (with a height range up to 1,500 m and measurement frequency of 16 seconds).





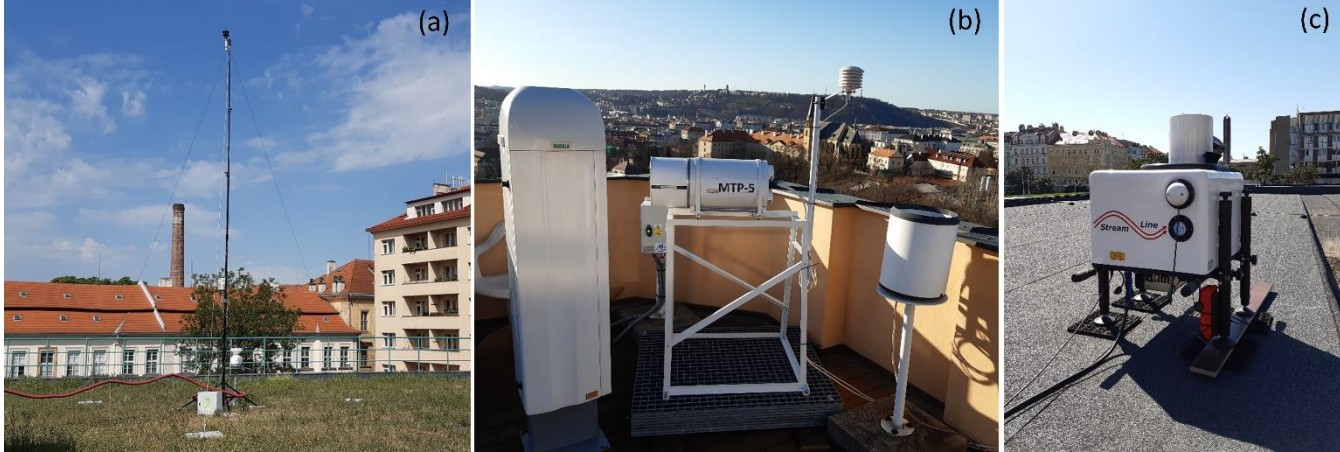

**Figure. 4**. Supplementary non-reference meteorological measurement used for TURBAN observation campaign: (a) mobile telescopic meteorological mast (MM, height 7.5 m) installed in PVK garden, (b) microwave radiometer MTP-5-He (MWR; for temperature profile measurement up to 1 km height) placed at the Karlov MS; (c) Doppler LIDAR StreamLine XR (for wind and backscatter intensity measurement up to 1.2 km height) placed at the PVK roof.

**2.3 Data processing and statistical analyses**

All data were quality checked before averaging, i.e. the missing values (caused by instrument defects, power outages etc) were marked as not available values (NAs). If fewer than 70 % of the data samples were available in a given hour, the entire hourly average was also marked as missing value (NA). For all data the UTC time was used, the timestamp always corresponds to the beginning of the averaging interval.

**2.3.1 Statistical tools used**

For the data processing and visualisation, R software (R Core Team, 2021) with the following R packages were used: ggplot2 (Wickham, 2016), corrplot (Wei and Simko, 2021), openair (Carslaw and Ropkins, 2019), tdr (Lamigueiro, 2022). Furthermore the TIBCO Statistica software (version 13.1.0; TIBCO, 2020) was used for calculation of the MARS correction equations for individual LCSs and for the basic summary statistics. Alternatively, there is also possible to use the R package earth 280 (Milborrow, 2011) for calculation of MARS corrections with R software. The interpolated meteorological profiles were visualised with the Golden Software Surfer (version 19.4.3; Surfer, 2022).

**2.3.2 LCS air quality data control and correction methods**

A summary diagram of the entire process of LCS air quality data control, applied correction methods and evaluation of correction performance is shown in Fig. 5.



Before the actual use of the air quality LCSs at the final deployment sites, a sufficiently long initial field comparative measurement of all LCSs was carried out on the rooftop of the Prague 4-Libuš AQM station. During this testing period, from 21 December 2021 until 30 May 2022 (165 days in total), defective sensors were identified (3 out of 20 in total, 2 were later replaced), the settings of all sensors were synchronised (device time and data transfer to the data storage server) and deviations in measurements were identified both between individual LCSs and between LCSs and RMs (gasses) or EM (aerosol).

The initial field comparative measurement showed that most of the LCSs (17 stations, except 3 broken ones) followed a similar course of concentrations over time; however, with very different biases in absolute concentrations (see Fig. 6). The average coefficients of variation (CVs) and its standard deviations (SDs) of 1-hour averaged raw concentrations measured by all LCSs were 27.69±7.58 % for $NO_2$, 16.71±2.62 % for $O_3$, 23.44±9.33 % for $PM_{10}$ and 23.16±9.94 % for $PM_{2.5}$. In comparison with RM or EM the results of linear regression with raw LCS data showed values of coefficients of determination ($R^2$) ranging

between 0.84-0.98 for $NO_2$, 0.54-0.82 for $O_3$, 0.72-0.89 for $PM_{10}$ and 0.85-0.91 for $PM_{2.5}$ (for the complete basic statistics of all LCSs including particular statistical errors MBE, MAE and RMSE see Tables S2-S5 and Figs. S1-S4 in the Supplement). Since no significant outliers (defined as values greater than three times the maximum hourly average concentration measured by RM or EM; Bauerová et al., 2020; van Zoest et al., 2018) were identified during this testing period (not even during the final deployment at Legerova or during the final field comparative measurement), all raw hourly average concentrations were

used for the further statistical correction process.

For the data consistency check and the chance of identifying possible random or systematic data drifts in raw LCS concentrations, the double mass curve (DMC) method was used (Searcy and Hardison, 1960). This method is based on the linear regression of cumulative 1-hour average concentrations measured by RM (as an independent variable; the abscissa) and the cumulative raw and corrected 1-hour average concentrations measured by particular LCSs (as a dependent variable; the

ordinate), both over the entire period. Any deviations identified from the linear regression fit then indicate a change or break points in LCS measurement (data gaps, abrupt or systematic gradual data drifts, change of measurement location). The DMC control in the case of raw LCS data gained during initial field comparative measurement showed no significant data drifts or deviations (see Fig. S5 in the Supplement).



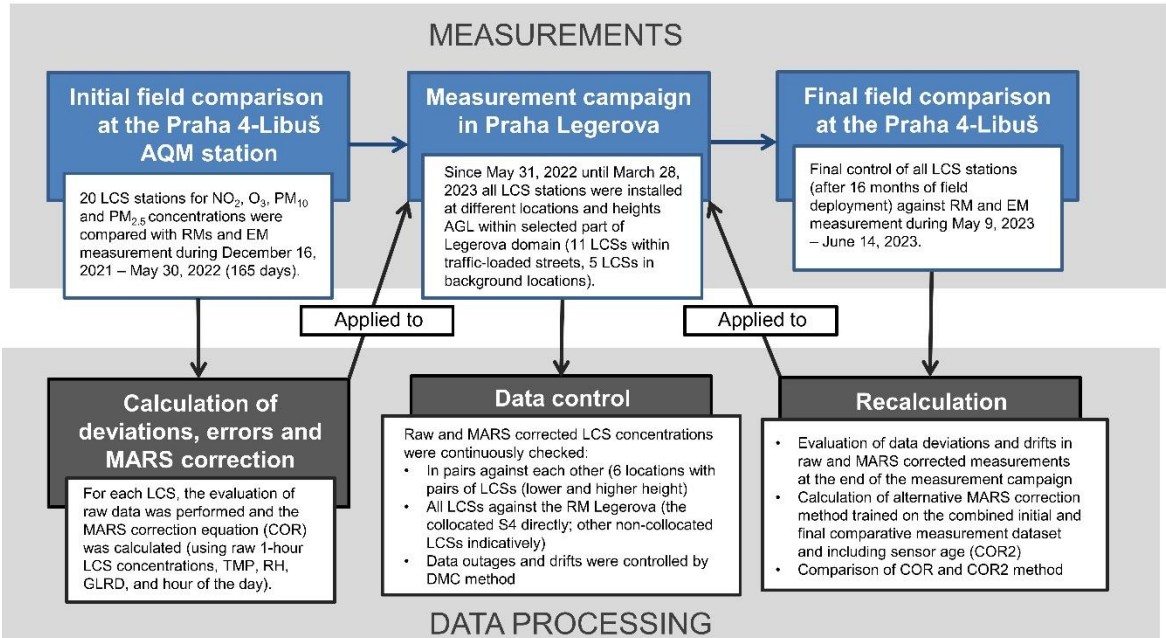

**Figure 5.** Summary scheme of particular steps in the entire process of LCS air quality measurement, data control, correction methods and evaluation of correction performance.

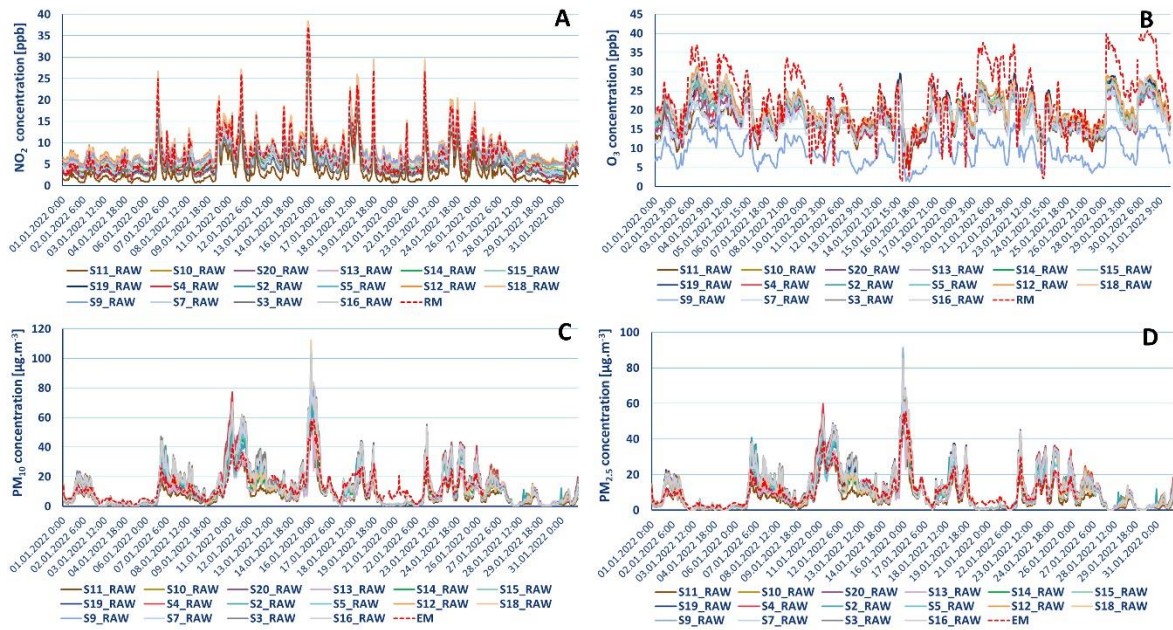

**Figure 6.** (a) Raw 1-hour average concentrations of $NO_2$, (b) $O_3$, (c) $PM_{10}$ and (d) $PM_{2.5}$ measured by all LCSs (marked as S2R, S3R, …S20R) in January 2022 during initial field comparative measurement at the Prague 4-Libuš AQM station (RM = reference monitor for gas measurement, EM = equivalent monitor for aerosol measurement).



For the mathematical correction of the LCS data the MARS method was used (Friedman, 1991). Correction equations (COR) were calculated for each LCS separately based on the whole training dataset gained during the initial field comparative measurement. The exception was the shortening of the training dataset in the case of sensors S3 (corrections based on dataset

from 16 December 2021 to 23 February 2022) and S4 (dataset from 16 December 2021 to 24 March 2022) because these were placed in the final deployment locations earlier than the other sensors. A different training period was also used in the case of sensor S8 (from 22 May 2022 to 31 January 2023), which was one of the broken LCSs at the start and was later returned to the Prague 4-Libuš AQM station and maintained as a spare LCS in the case of the failure of another one (since 15 February 2023 installed in place of the broken LCS S18 at the school Sokolská higher location).

All MARS corrections were built of 1-hour average concentrations measured by RM or EM as dependent variables and the following list of continuous independent variables: 1-hour average concentrations measured by an individual LCS, further 1-hour averaged TMP, RH, WV, GLRD and hour of the day. The maximum number of basis functions was set to 21, the degree of interactions to 1 (i.e. no interactions included), the penalty to 2, the threshold to 0.0005 and pruning was allowed. In the case of $O_3$ measurement, the raw $O_3/NO_2$ concentration and the ratio of $O_3/NO_2$ and $NO_2$ concentration from separate LCSs

were used as explanatory variables (for the possibility of taking into account the interference effect of the combined $O_3/NO_2$ sensor). The correction decreased the differences in the medians and ranges of concentrations between LCSs and RM or EM and between individual LCSs (see Fig. 7 and Fig. S6 in the Supplement). The average CVs and its SDs of all LCS concentrations were: 9.25±7.11 % for $NO_2$, 6.06±4.90 % for $O_3$, 13.05±15.29 % for $PM_{10}$ and 14.62±15.42 % for $PM_{2.5}$. The corrections also improved the relationship of the LCS data with the data from RM or EM: $R^2$ ranged between 0.89-0.99 for

$NO_2$, 0.91-0.96 for $O_3$, 0.75-0.92 for $PM_{10}$ and 0.91-0.95 for $PM_{2.5}$ (see Figures. S1-S4 and summary statistics including MBE, MAE and RMSE in Tables S2-S5 in the Supplement). The complete statistics of MARS corrections performance of each LCS, including the frequencies of use of each independent variable are listed in Tables S6-S13. Examples of correction equations for $NO_2$, $O_3$, $PM_{10}$ and $PM_{2.5}$ in the case of the LCS S2 are shown in Table S14. The improvement after the application of the MARS correction was also confirmed by the DMC method (see Fig. S7). However, it is important to mention that after

correction, some initially very low concentrations turned into weakly negative values: for gaseous pollutants they constituted less than 0.3 % and for aerosol less than 2.6 % of the whole testing dataset (part of the summary statistics in Tables S2-S5). Within the framework of testing different correction methods, one alternative method, hereinafter named COR2, was chosen. The COR2 method was specified as the MARS correction calculated on the basis of the combined dataset from the initial and final comparative measurements at the Prague 4-Libuš station, including the age of the LCSs (in days from the start of the

measurement) into the explanatory variables. However, because this correction method achieved very similar results to the original correction method (COR) based only on the initial field comparative measurement, it was not finally applied to the data measured in the Legerova campaign. For a description of the COR2 method and example of its results, please see Sect. S2.2.1 and Fig. S8 in the Supplement.





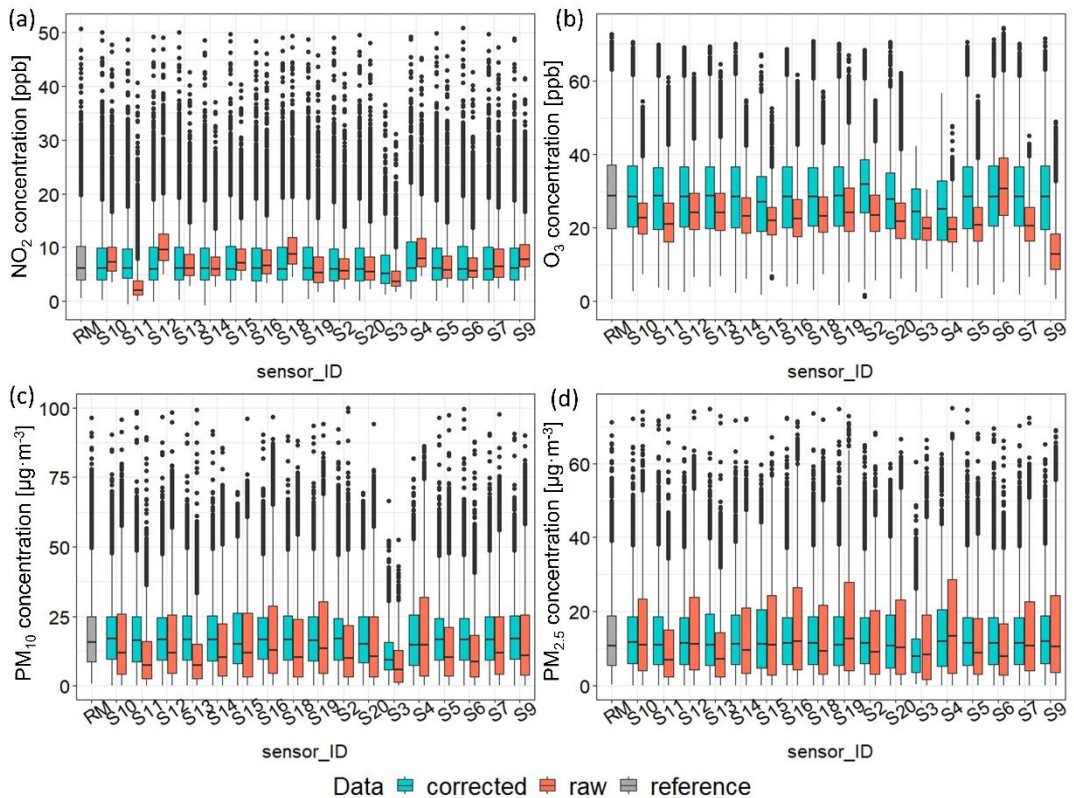

**Figure 7.** Boxplots showing medians and ranges of (a) NO$_2$, (b) O$_3$, (c) PM$_{10}$ and (d) PM$_{2.5}$ hourly averaged concentrations originally measured by LCSs (raw; red colour), corrected by the MARS method (corrected; blue colour) and by reference or equivalent method (RM; grey colour) during the initial field comparative measurement at the Praha 4-Libuš from 16 December 2021 to 30 May 2022.

The correction equations obtained for individual LCSs based on the initial testing dataset at the Prague 4-Libuš station (COR) were applied to the concentrations measured by particular LCSs during the Legerova measurement campaign using the necessary meteorological data from the MS Prague Karlov.

The first stage of the possible data check within the Legerova campaign was a mutual comparison of data from LCSs installed in pairs at the same locations but at different height levels above the ground (i.e. S11+S10, S20+S13, S14+S15, S2+S5, S12+S18, S9+S7, always mentioned as lower+higher height level) and the LCS S4 collocated with the RM Legerova for entire period (see map in Fig. 1b). In all cases (NO$_2$, O$_3$, PM$_{10}$ and PM$_{2.5}$), the concentrations measured within particular pairs of LCSs were in most cases highly correlated to each other (see Figures S9-S16 with courses of concentrations and Figures S17-S18 in Supplement showing particular relationships with the RM Legerova measurement). The only sensor identified as defective was the NO$_2$ LCS S9 placed within the closed school courtyard, which showed gradually increasing data drift to high concentrations over time (Fig. S14a in the Supplement).



The possible data drifts during the entire measurement period were further checked with the DMC method. For the Legerova campaign, all the LCS raw and corrected concentrations were indicatively compared with the concentration measured with RM or EM at the Legerova AQM station. In the case of $NO_2$, the previously mentioned LCSs S9 and further S11 and S12 were identified with an indication of a systematic gradual data drift (see Fig. S19). Some technical issue must have occurred in the $O_3$ LCSs, where in all sensor units a sudden partial data drift occurred in October to November 2022 (see Fig. S20). Therefore there is no clear warranty in $O_3$ data after 15 October 2022. In the case of the aerosol LCS measurement ($PM_{10}$ and $PM_{2.5}$) no data drifts were identified based on DMC (Fig. S21). The complete results are shown in Sect. S2.3.2 in the Supplement.

At the final stage, after the end of the Legerova measurement campaign (finished on 28 March 2023), all LCSs were uninstalled and moved back to the Prague 4-Libuš AQM station for the final field comparative measurement lasting from 9 May 2023 to 14 June 2023 (37 days). The ranges and medians of raw and MARS-corrected LCS concentrations of all pollutants are shown in Fig. S22 in the Supplement. In the case of $NO_2$ measurement, 10 out of 17 LCSs achieved $R^2 > 0.80$ with corrected concentrations (COR), 5 LCSs achieved $R^2 > 0.60$, LCS S4 achieved $R^2 = 0.56$ and the weakest relationship was detected in S9 with $R^2 = 0.17$ (for the particular statistics of all LCSs including statistical errors see Table S15 and Fig. S23 in the Supplement). The absence of a relationship in the case of LCS S9 during the final comparison confirmed the sensor failure and therefore this sensor was not further used for evaluation. In the case of $O_3$ the improvement of the relationship between RM and the corrected data was not significant at the end, in LCSs S3 and S4 the relationships were even slightly worsened (Fig. S24). Although $R^2 > 0.85$ was achieved in all $O_3$ LCSs compared to RM, the intercept was shifted to negative values (i.e. the concentrations after MARS correction were underestimated at the end of measurement campaign in most of the $O_3$ LCSs; for complete statistics see Table S16 and Fig. S24). In aerosol measurement the weakest relationships in comparison with EM were reached in the case of $PM_{10}$ concentrations. Although MARS-corrected concentrations significantly improved the relationship with EM and narrowed the variation of originally measured concentrations even at the end of the campaign, the $R^2$ ranged only from 0.47 to 0.63 (Table S17 and Fig. S25). The worst relationship was achieved in the case of the S3 sensor, which, even after correction, significantly underestimated the $PM_{10}$ concentrations compared to the EM (see Fig. S25). Better relationships were achieved in the case of $PM_{2.5}$ measurement with $R^2$ values between 0.73 and 0.89, where none of the LCSs achieved significantly underestimated or overestimated concentrations even at the end of the campaign (see Table S18 and Fig. S26).

**2.3.2 Data quality of supplementary meteorological measurement**

All 10-minute data measured by the non-reference mobile mast (MM) at the PVK garden, namely TMP, RH, p, WD and WV were compared to the referential data measured at the adjacent Prague Karlov MS (337 m aerial distance; 20 m altitude difference) during the Legerova campaign from 1 June 2022 to 19 April 2023 (322 days in total, n = 46389). In the case of TMP, RH and p, the values of $R^2$ were higher than 0.98 (Fig. S27 in the Supplement). The biggest differences were detected in the case of WD and WV measurements (Fig. S27), which is understandable because the wind measurement in the PVK garden location was influenced by the surrounding building blocks (unlike the wind measurement at the Prague Karlov MS





located on the roof of the tallest building). Finally, 1-hour averages were calculated from all variables measured by MM in the PVK garden.

The vertical profiles of TMP from the MWR were checked gainst TMP vertical profiles measured by radiosonde launched

from the Prague Libuš MS during the period from 25 February 2022 to 24 March 2023 (392 days in total, n = 1172). The 5-minute TMP data measured by the MWR and corresponding radiosonde data at selected heights above ground (0, 50, 100, 500, 750 and 1,000 m AGL) and selected times (times of radiosonde launching at 0, 6 and 12 UTC) were used for comparison. Overall, the data showed very good agreement with $R^2 > 0.98$ even at the highest level of 1,000 m AGL (Fig. S28). For results of comparisons for particular sounding times under different conditions (including days with precipitation), please see Figures

S28-S30 in the Supplement. The correctness of the potential TMP profile calculation procedure was also verified on the TMP height profile from the radiosonde output. The resulting difference between the calculated and measured potential TMP did not exceed in absolute value 0.137 % of the value determined according to Arya (2001) and in comparison reached $R^2 = 0.997$ (see Fig. S31).

In the case of the Doppler LIDAR, the processed wind profile data (producing WV and WD at particular heights) from the

VAD 6 scanning program were captured roughly every 33 minutes. The actual comparison of the measured wind profile data with the reference method or the radiosonde data was not carried out within this study (in the case of wind profiles unsuitable for comparison due to the greater distance between the stations Prague 4-Libuš and Prague 2-Karlov). Additionally, for possible future uses of turbulent kinetic energy (TKE) assessment above the selected domain in Prague, the TKE scan in the CSM regime with the elevation angle 35.26° lasting 30 minutes in a total of 25 cycles was set to obtain the course of the radial wind

component (Vr). To calculate the resulting value of TKE according to the Smalikho and Banakh  (2017) method the standard deviation of Vr should be calculated for each range gate and each azimuth (from 25 values) and subsequently averaged over all azimuths. For detecting the maximum height of valuable wind profiles measured by the Doppler LIDAR (according to SNR values), two possible methods were tested. The first method was based on cutting the profile at a certain SNR threshold (i.e. cutting of the values with SNR>1.015 like in Tzadok et al., 2022); see the example in Fig. S32). In the second method the

standard deviation (SD) of the WV was calculated in the sliding high range window and subsequently the DMC method for flexible identification of height where a sudden jump of the SD occurred has been utilised (see complete method description in Sect. S2.4.3 and example in Fig. S33 in the Supplement).

## 3 Results

### 3.1 Air quality monitoring within Legerova campaign

The difference between raw measured and MARS-corrected $NO_2$, $O_3$, $PM_{10}$ and $PM_{2.5}$ concentrations from each LCS is shown in Fig. 8. In the case of $NO_2$, which is one of the primary emission outputs from transport, the results showed a significant difference in the concentration trends measured during working days (with a high traffic intensity in the monitored streets) and during the weekends (when automotive traffic is decreased; Fig. 9). Furthermore, the effect of the morning (from 6 a.m. to 9





a.m. UTC) and afternoon (from 3 p.m. to 6 p.m. UTC) rush hours was clearly visible during working days (Fig. 9). The highest

1-hour average concentrations were measured by the most exposed LCSs during August 2022 and November 2022 (Fig. 9).

Given the medians and even the averages of 1-hour $NO_2$ concentrations, the most exposed locations were: CKAIT Sokolská

(at the crossroads of Sokolská and Rumunská streets) with the LCSs S10 measuring a median concentration of 33.21 ppb at

the higher height and the S11 measuring a median of 31.12 ppb at the lower height; Legerova (at the crossroads of Legerova

and Rumunská streets) with the LCSs S14 at the lower height and the S15 at the higher height, both with a median concentration

of 25.13 ppb; and Rumunská with the LCS S20 measuring a median concentration of 24.49 ppb at the lower height and S13

measuring a median of 23.34 ppb at the higher height (see Fig. 8a, Fig. 9 and Fig. 11a, Table S19 in the Supplement). The

maximum 1-hour average $NO_2$ concentrations were 129.93 ppb measured by the LCS S12 (Sokolská school at lower height)

and 92.50 ppb measured by the LCS S18 (Sokolská school at higher height; Fig. 11b and Table S19). According to the median

or average $NO_2$ concentrations, the Sokolská school location was rather moderately polluted, similarly to the nearby Legerova

school location (LCSs S2 and S5; both sites with the median concentrations ranging between 18.58 and 20.35 ppb) and the

Prague Legerova RM with the collocated LCS S4 (with median concentrations 18.82 and 20.67 ppb, in the given order; Fig.

8a, Fig. 9 and Fig. 11a, Table S19). The lowest $NO_2$ 1-hour average concentration were measured with the background LCSs,

namely the S3 placed on the roof of the Prague Karlov MS and the S16 placed on the roof of the Le Palais Art Hotel Prague

(concentrations < 10 ppb), and further with LCSs S19 placed at the PVK garden (concentration 11.28 ppb) and S7 placed

within the closed school courtyard (concentration 11.47 ppb; Fig. 8a, Fig. 9 and Fig. 11a, Table S19). The background LCS

S9 (school courtyard, lower height) was not included in the evaluation due to the detected significant data drift during the

observation campaign.

In the case of $O_3$ LCS measurement no significant change was detected in the course of concentrations between weekdays and

weekends (Fig. S34 in the Supplement). The highest $O_3$ concentrations were measured around midday (from 11 a.m. to 2 p.m.)

and quite understandably during the summer months (from June until August 2022; Fig. S34). The difference in the LCS-

measured $O_3$ concentrations probably depended strongly on the individual conditions of particular locations. The highest

medians of average 1-hour $O_3$ concentrations were 13-16 ppb measured by the LCSs S7, S18, S11, S16 (Fig. 8b and Fig. S34)

and the maximum concentrations were 103-109 ppb measured by the LCSs S14, S20, S7 and S9 (see complete statistics in

Table S20 in the Supplement). Since there is no RM for measuring $O_3$ available at the Prague 2-Legerova AQM, the data from

the Prague 9-Vysočany RM were used for indicative comparison with all LCS measurements at the Prague Legerova domain.

In the case of aerosol particle pollution, the measurement also showed a difference between weekdays and weekends. The

highest 1-hour average $PM_{10}$ and $PM_{2.5}$ concentrations were measured on Wednesdays, Thursdays and surprisingly also

Sundays, while a significant drop in aerosol concentrations was detected on Saturdays (for $PM_{10}$ see Fig. 10 and for $PM_{2.5}$ see

Fig. S35 in the Supplement). This was probably partly a response to the change in the traffic regime and partly to other local

emission sources (household heating during winter; see the highest concentrations during the winter months in Fig. 10).

However, in general, no extremely high levels of $PM_{10}$ or $PM_{2.5}$ pollution were detected within the entire area of interest,

despite the high traffic load in the monitored streets. LCSs S10 and S11 placed in Sokolská street (at the crossroads with





Rumunská), LCSs S14 and S15 in Legerova street (at the crossroads with Rumunská) and LCSs S20 and S13 in Rumunská
street were again the locations with the highest medians of 1-hour average $PM_{10}$ and $PM_{2.5}$ concentrations ranging between
23-26 µg·m⁻³ and 15-18 µg·m⁻³ (respectively; see Fig. 8 and Fig. 10, Fig. S35 and Tables S21-S22). On the contrary, the LCSs
placed on the school building at the exit to the Nuselské valley and the LCSs placed in the background locations were less
loaded (similarly as in the case of $NO_2$ pollution), with medians of measured $PM_{10}$ concentrations <20 µg·m⁻³ (Fig. 8 and Fig.
10, Fig. S35, Tables S21-S22). The lowest average concentrations were in the case of $PM_{10}$ and $PM_{2.5}$ measured by LCS S3
placed on the roof of the Prague Karlov MS (median of $PM_{10}$ concentration 11.41 µg·m⁻³, $PM_{2.5}$ concentration 9.14 µg·m⁻³).
The maximum 1-hour average $PM_{10}$ and $PM_{2.5}$ concentrations were achieved by LCSs S5, S13, S10, RM and S2 (for complete
statistics see Tables S21-S22) and were significantly influenced by the temporary pollution episode in July 2022 (see Sect.
3.2). The medians and maxima of $PM_{10}$ concentrations measured during the entire measurement campaign at different locations
are shown in maps in Fig. 8c and Fig. 8d.

With the correction method COR2 (also taking into account the final comparative measurement and the age of the LCSs), very
similar results were achieved, with the difference that COR2 occasionally performed inappropriately at low concentrations
(shift in intercept/absolute term). The example of linear regression results of $NO_2$, $O_3$ and $PM_{10}$ concentrations corrected by
COR and COR2 method in the case of LCSs S2, S4 and S6 are shown in Fig. S36 in the Supplement.

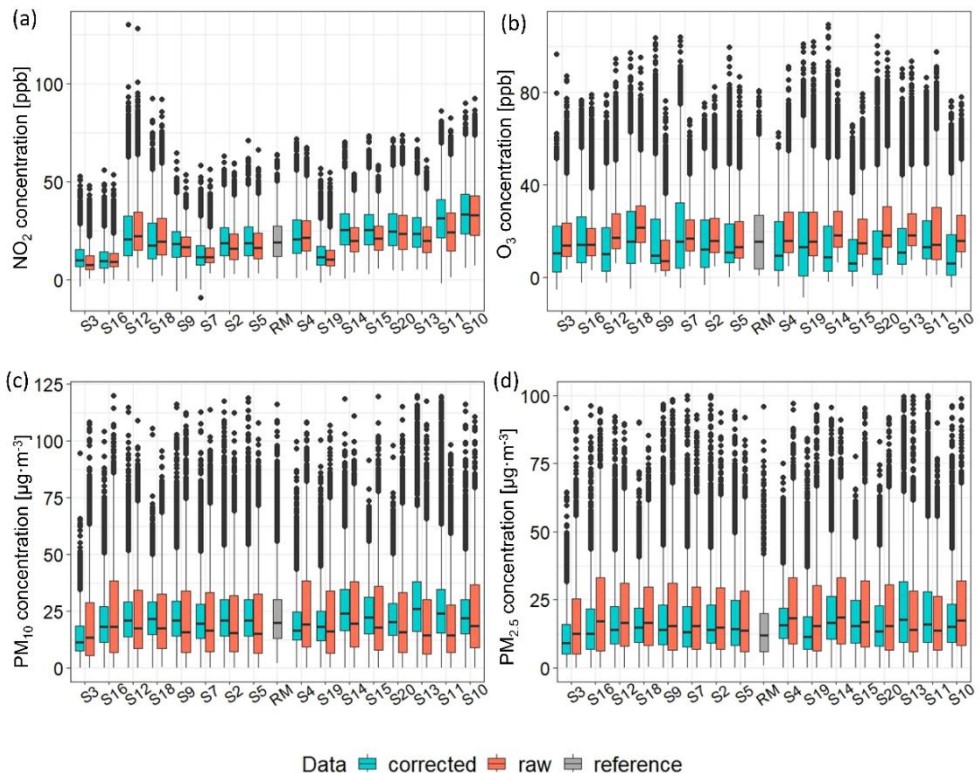

**Figure 8.** Boxplot showing medians and ranges of (a) $NO_2$, (b) $O_3$, (c) $PM_{10}$ and (d) $PM_{2.5}$ hourly averaged concentrations originally
measured by LCSs (raw; red colour), corrected by the MARS method (corrected; blue colour) and by reference or equivalent method (RM;





grey colour) during the Praha Legerova measurement campaign lasting from 30 May 2022 to 28 March 2023. The X-axis is sorted according to the measurement deployment sites.

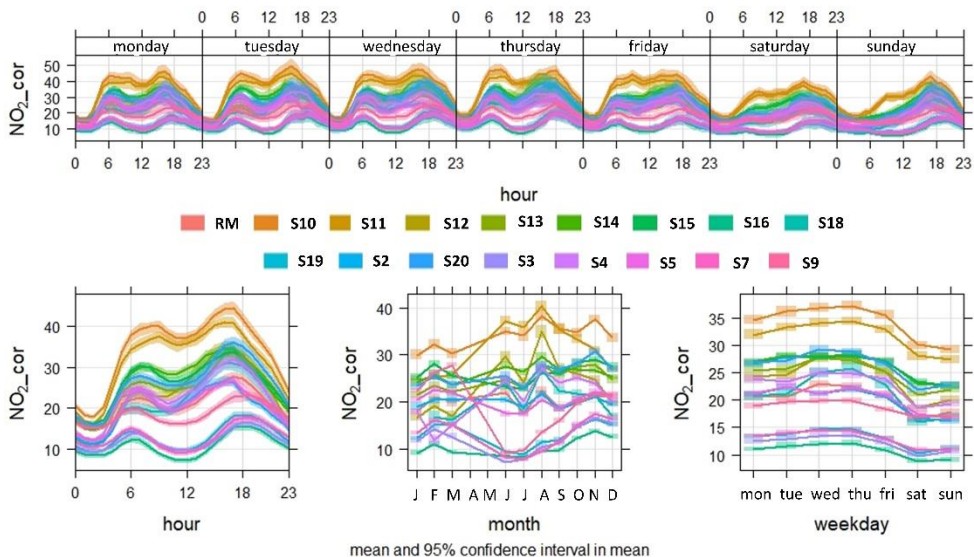

**Figure 9.** Daily (top), hourly (bottom left), monthly (bottom middle) and weekly (bottom right) of corrected $NO_2$ concentrations (ppb) measured by all low-cost sensor stations (LCSs S2-S20) and by the Praha 2-Legerova reference monitor (RM) within the Legerova campaign. Measuring period from 30 May 2022 to 28 March 2023 (in monthly graph May to December 2022, January to March 2023).

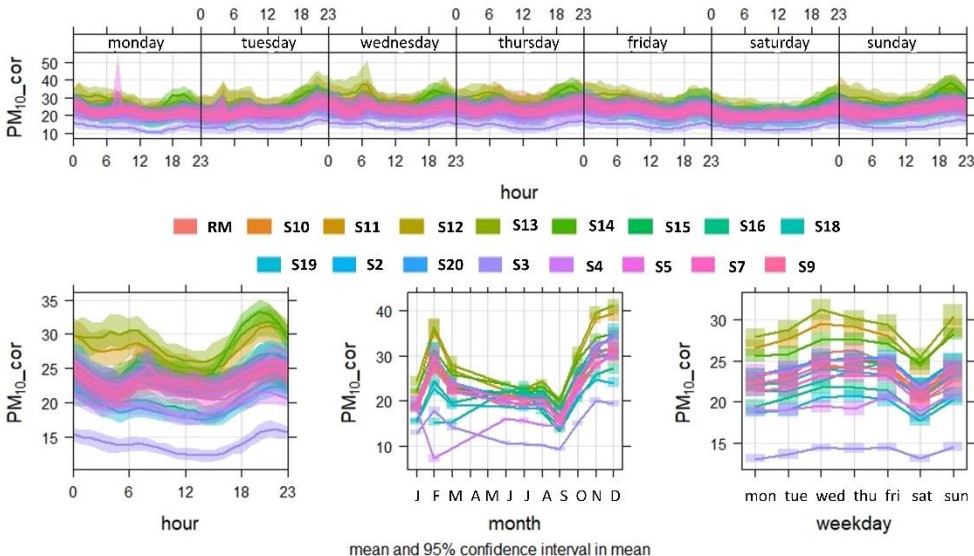

**Figure 10.** Daily (top), hourly (bottom left), monthly (bottom middle) and weekly (bottom right) variations of corrected $PM_{10}$ concentrations ($\mu g \cdot m^{-3}$) measured by all low-cost sensor stations (LCSs S2–S20) and by equivalent monitor the Praha 2-Legerova within the Legerova campaign. Measuring period from 30 May 2022 to 28 March 2023 (in monthly graph May to December 2022, January to March 2023).





**Figure 11.** Map with measurement locations showing: (a) medians and (b) maximum of NO$_2$ concentrations (ppb); (c) medians and (d) maximum of PM$_{10}$ concentrations (µg·m$^{-3}$). Both measured during the entire measurement period (from 30 May 2022 to 28 March 2023) in Legerova and its surroundings. The sensors were placed at two height levels in six locations (see legend). The colour scales differ between medians and maximum concentrations and between pollutants. Background map is provided through WMS by the Czech Office for Surveying, Mapping and Cadastre – ČÚZK.



## 3.2 Episodes with temporarily increased air pollution concentrations

A significant pollution episode was recorded in July 2022, when a large-scale forest fire broke out in the České Švýcarsko National Park (the northern part of the Czech Republic, see in Fig. 1a) and the aerosol pollution emitted into the air spread across the republic over long distances. On 26 July 2022 around 4 a.m. and 9 p.m. (both UTC), this transported aerosol pollution was also detected in Prague. The entire LCS network (including background locations) reacted very well with the significant increase in $PM_{10}$ and $PM_{2.5}$ concentrations (Fig. 12). This aerosol pollution was also detected by increased backscatter intensities from the CL51 ceilometer at Prague Karlov and from the Doppler LIDAR placed at the PVK garden (see Fig. S37 and S38 in the Supplement).

Some temporary episodes with high concentrations occurred mainly in aerosol pollution. Increased concentrations of $PM_{2.5}$ and $PM_{10}$ were measured during the temperature inversions, when disperse conditions were worsened and negative values of the TMP gradient were detected from the MWR measurement (see Fig. 13 with $PM_{2.5}$ concentrations over the whole measurement period and Fig. 14 with examples of $PM_{10}$ episodes during September and December 2022 and February 2023). Similarly, high concentrations of $PM_{10}$ and $PM_{2.5}$ were measured during New Year's Eve (see Figs. S39 and Fig. S40 in the Supplement).

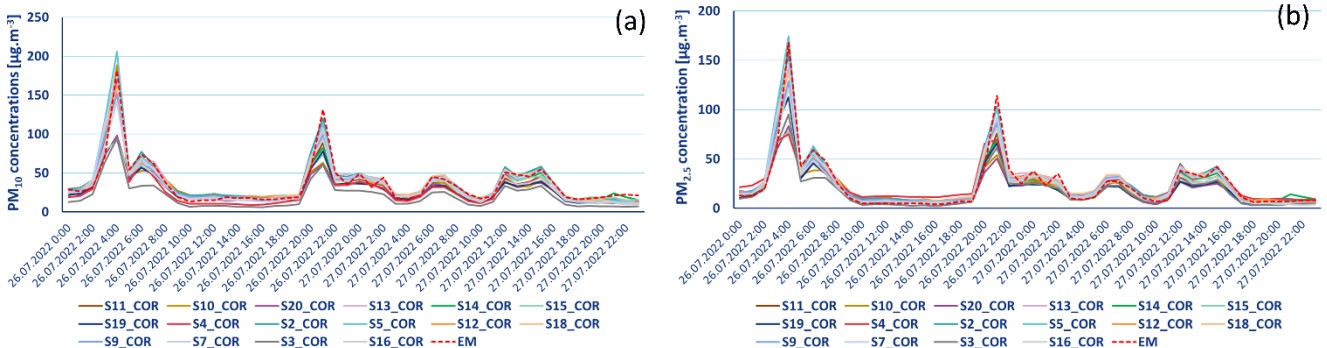

**Figure 12.** Concentrations of (a) $PM_{10}$ and (b) $PM_{2.5}$ measured by LCS TURBAN network and Fidas equivalent monitor (EM) at Legerova AQM station during the pollution episode caused by aerosol transported from a large-scale forest fire in Hřensko (the northern part of the Czech Republic) on 26 July 2022 in the morning and evening hours.



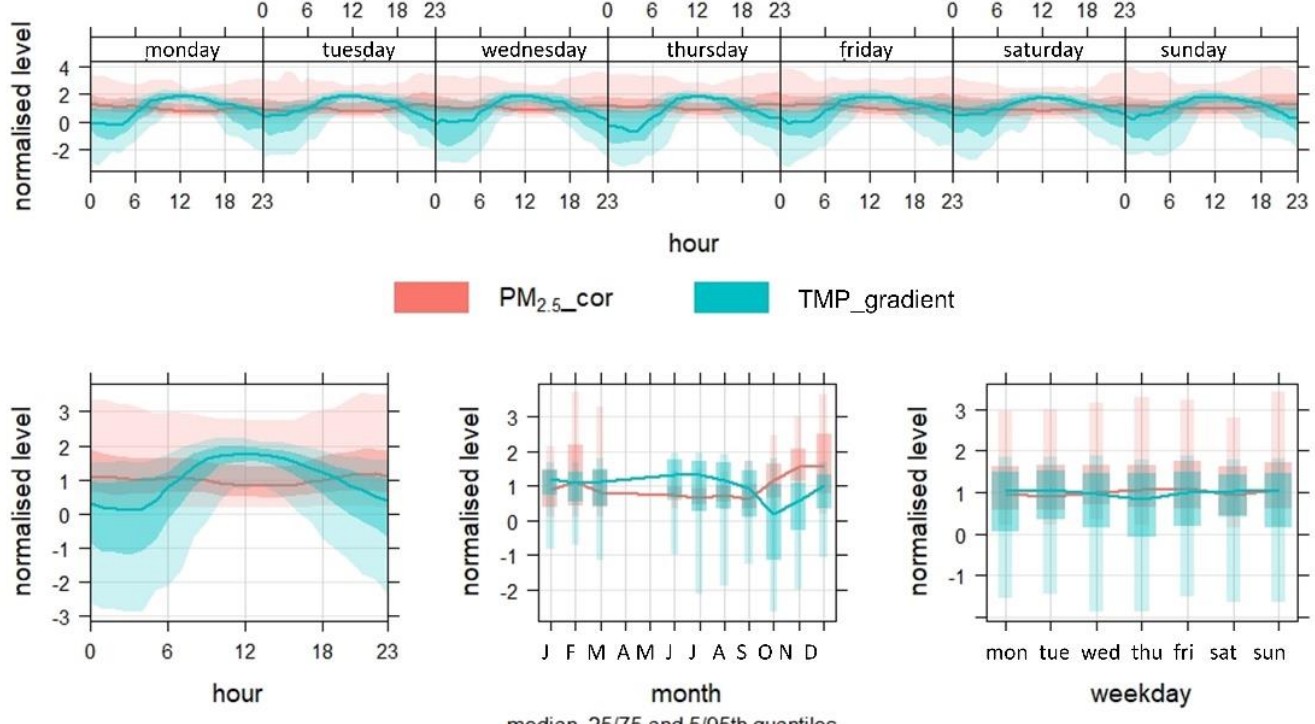

**Figure 13.** Daily (top), hourly (bottom left), monthly (bottom middle) and weekly (bottom right) variations of corrected PM$_{2.5}$ concentrations (µg·m$^{-3}$; hourly averages from all LCSs) and TMP gradient (°C/100 m), both variables normalised for comparison. The median and quantiles are shown during the whole Legerova measurement campaign from 30 May 2022 to 28 March 2023 (in the monthly graph May to December 2022, January to March 2023).







**Figure 14.** The course of PM$_{10}$ concentrations (μg·m$^{-3}$) during 22-30 September 2022 (top), 8-18 December 2022 (middle) and 6-17 February 2023 (bottom). An increase in PM$_{10}$ concentrations is evident under conditions of ground temperature inversion (shown as negative temperature gradient, TMP_gradient).

525



## 3.3 Meteorological measurement in Legerova domain

The results of MM measurement at the PVK garden between 1 June 2022 and 19 April 2023 showed quite normal (expected) courses of TMP, RH and WV (Figs. S41-S43 and summary statistics in Tables S23-S25 in the Supplement). In the case of TMP, the threshold of 30 °C was exceeded for a total of 11 days during June and July 2022 (maximum TMP 37.13 °C on 19 June 2022; the longest period of TMPs over 30 °C lasted 4 consecutive days during 19-22 July 2022). Conversely, TMPs below 0 °C were measured in 9 consecutive days during 10-19 December 2022. The coldest 1-hour average TMP of −8.88 °C was observed during the cold period of 12-14 December 2022. The TMP gradients calculated from the lowest 200 m of TMP profiles showed that in the period between 23 February 2022 and 28 March 2023 (398 days) there were a total of 279 days with the occurrence of TMP inversion conditions. The deepest inversion (TMP gradient >−3.5 °C/100 m) was detected on 24 March 2022 between 3 a.m. and 5 a.m. UTC (Fig. S44 in the Supplement).

Here we present an example with a fast reconstruction of TMP stratification in the atmospheric boundary layer during 24 September 2022 (see the change of atmospheric stability according to the potential TMP gradient in Fig. 15). On this day the vertical profile measurements showed the TMP inversion with a peak at the height of 400-450 m AGL at 3 and 6 a.m. UTC (with the TMP difference of 2.56 °C between 0 and 400 m height AGL at 3 a.m. UTC and 2.76 °C between 0 and 450 m height AGL at 6 a.m. UTC; see Fig. 16a). At the same times (3 and 6 a.m. UTC), two layers with evident low-level jet were noted in the wind profile within this nocturnal inversion (at height 46.6 m AGL WV around 1.5 m·s⁻¹ and at 226 m AGL 6.49 m·s⁻¹ at 3 a.m. UTC and at 173 m AGL 4.58 m·s⁻¹ at 6 a.m. UTC with a rapid decrease of the WV with height in both cases) including a partial change in WD between 46.4 m and 200 m AGL (see Fig. 16b and Fig. 16c). At 8 a.m. UTC, the TMP inversion was no longer occurring and at 3 p.m. UTC the profile was already almost adiabatic (see Fig. 16a and Fig. 16d). During non-inversion conditions, the WV and WD were much more variable in lower heights (i.e. between 46.4 m and 100 m AGL; see 8 a.m. and 3 p.m. UTC cases in Fig. 16e - Fig. 16f). Then at 10 p.m. UTC, the nocturnal temperature inversion was again noted (Fig. 16d). This change in atmospheric stratification corresponded well with the pollution situation, especially in the case of aerosol (NO₂ concentrations followed the traffic regime in the streets more; see Fig. 17). Examples of slower and less intense change in stratification including low-level jets and the follow-up to the air pollution changes is shown on 13 February 2023 in Fig. 18 and Fig. 19. Further examples are given in Fig. S45 – Fig. S48 in the Supplement.



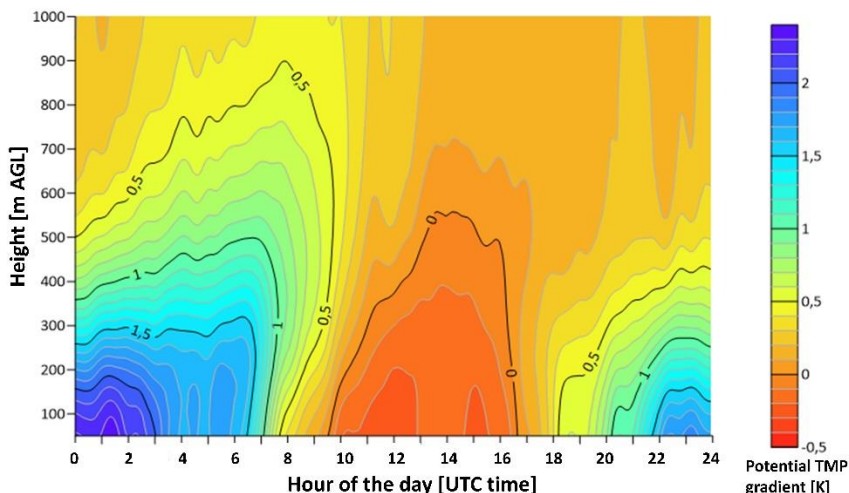

**Figure 15.** Evolution of the stability of the atmospheric boundary layer according to the potential temperature gradient measured by MWR on the Prague 2-Karlov MS roof on 24 September 2022.

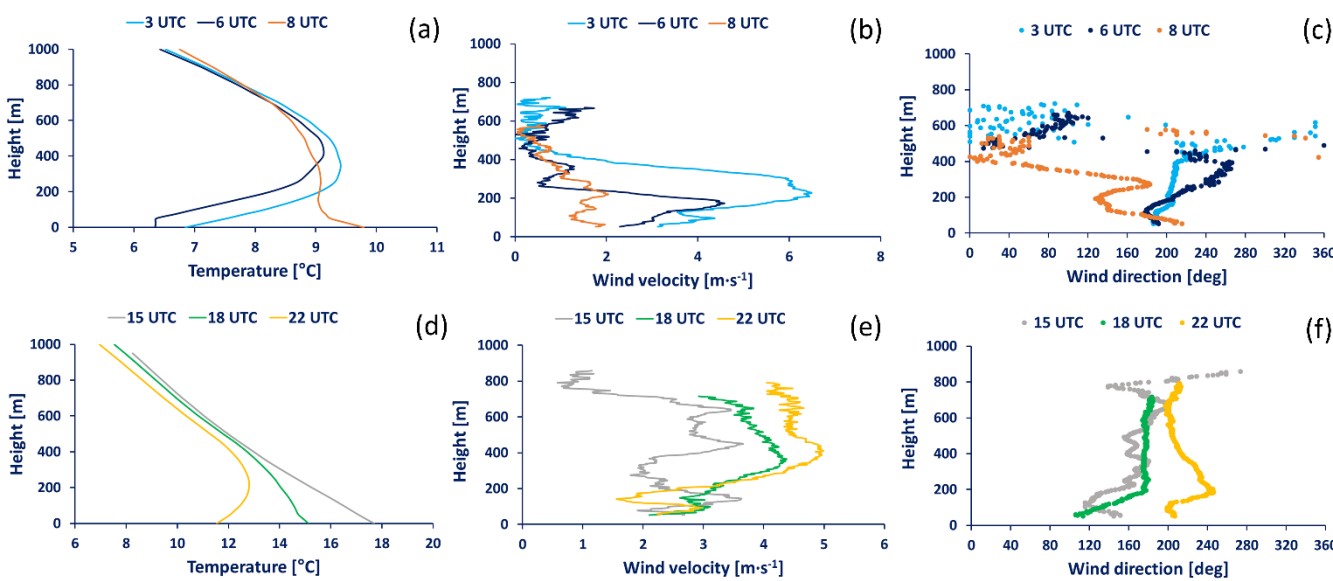

**Figure 16.** Example of temperature (TMP), wind velocity (WV) and wind direction (WD) profiles during 24 September 2022 with a fast reconstruction of TMP stratification within the lower boundary layer. The TMP profiles were measured by MWR at the Prague Karlov MS at (a) 3:00, 6:00 and 8:00 UTC time and at (d) 15:00, 18:00 and 22:00 UTC time; WV and WD profiles were measured by LIDAR at the PVK roof at (b+c) 3:00, 6:00 and 8:00 UTC time and at (e+f) 15:00, 18:00 and 22:00 UTC time.



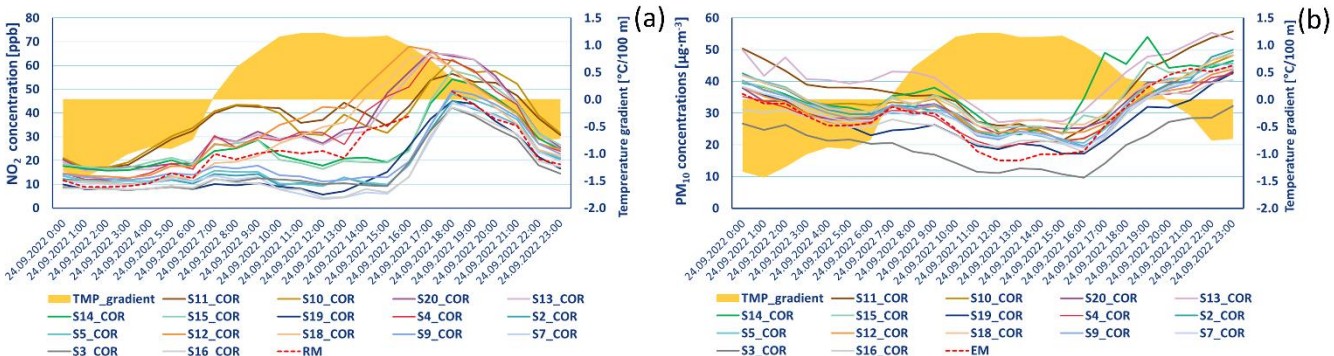

**Figure 17.** The course of 1-hour average (a) $NO_2$ and (b) $PM_{10}$ concentrations measured by low-cost sensors at different locations within the Legerova campaign on 24 September 2022. The temperature inversion and non-inversion conditions are shown by the TMP gradient (°C/100 m).

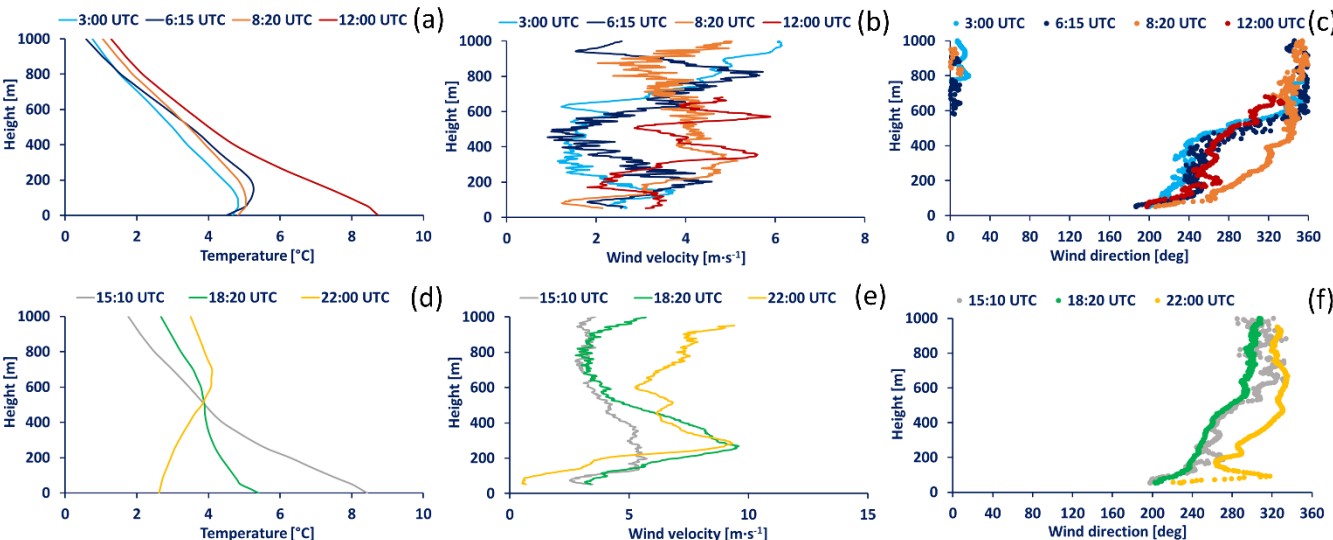

**Figure 18.** Example of temperature (TMP), wind velocity (WV) and wind direction (WD) profiles during 13 February 2023 with change in TMP stratification within the lower boundary layer. The TMP profiles were measured by MWR at the Prague Karlov MS at (a) 3:00, 6:15, 8:20, 12:00 UTC and at (d) 15:10, 18:20 and 22:00 UTC; WV and WD profiles measured by LIDAR at the PVK roof at (b+c) 3:00, 6:15, 8:20, 12:00 UTC, and at (e+f) 15:10, 18:20 and 22:00 UTC.



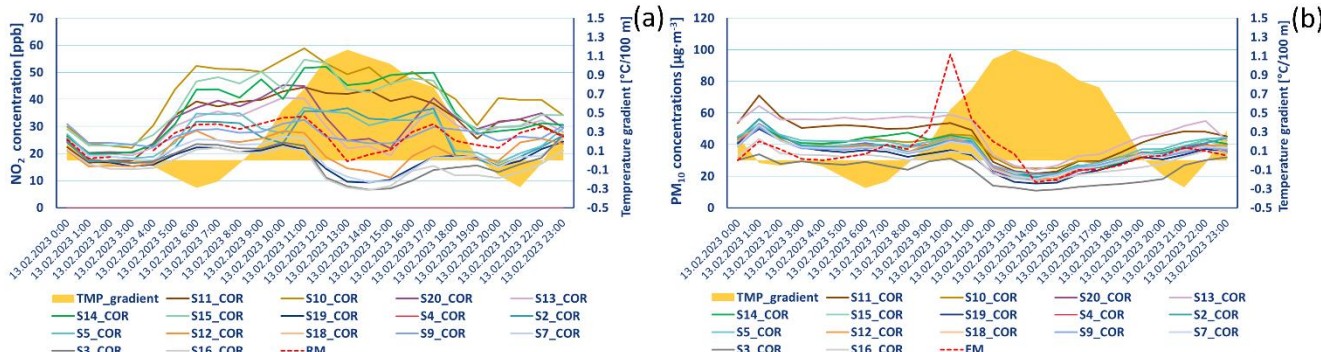

**Figure 19.** The course of 1-hour average (a) $NO_2$ and (b) $PM_{10}$ concentrations measured by low-cost sensors at different locations within the Legerova campaign on 13 February 2023. The temperature inversion and non-inversion conditions are shown by the TMP gradient (°C/100 m).

## 4 Discussion

The discussion is structured with respect to the sub-topics addressed in this article.

### 4.1 Data quality of raw and MARS-corrected LCSs air quality measurement

#### 4.1.1 Raw LCS measurement

With regard to the set study design based on the long-term initial field testing of all LCSs at the Prague 4-Libuš RM station (in total lasting 5.5 months), it was not sure whether all LCS units (especially the EC Cairsens $NO_2$ and $O_3$ LCSs with a stated maximum operational life of 15 months) would be able to measure without failure during the entire Prague Legerova measurement campaign. Finally, no major data outages or LCS malfunctions occurred with the exception of three EC LCSs which were identified as defective at the beginning of the test period and two LCS stations where the communication unit failed during the Legerova measurement campaign (namely the LCS S18 located at Sokolská school had been broken since 13 December 2022, and S4 located at RM Legerova had been broken since 5 February 2023; both were subsequently corrected and returned to the final field comparative measurement at the Prague 4-Libuš RM).

Evaluation of raw LCS measurement showed a quite high correlation with RM in the case of $NO_2$ Cairsens LCSs ($R^2>0.84$) and with EM in the case of $PM_{10}$ and $PM_{2.5}$ Plantower LCSs ($R^2>0.72$ and $R^2>0.85$). The weakest correlation was detected for combined $O_3/NO_2$ Cairsens LCSs where two units reached only $R^2=0.52$ and three units $R^2=0.69$ in comparison with $O_3$ RM (the other units with $R^2>0.76$). However, all LCSs suffered from varying zero shift (intercept shift), which resulted in the following ranges of MBE: −2.98-4.56 ppb in $NO_2$, −3.41-14.52 ppb in $O_3$, −3.54-8.16 µg·m$^{-3}$ in $PM_{10}$ and −3.92-3.90 µg·m$^{-3}$ in $PM_{2.5}$. Especially in the case of EC Cairsens sensors, we achieved better results in raw measurements than in some older studies using similar sensor types in outdoor conditions (with maximum $R^2$ varying between 0.50 and 0.80 or even lower; see (Bauerová et al., 2020; Feinberg et al., 2018; Jiao et al., 2016; Spinelle et al., 2015). From this point of view, some relevant





technical improvements could have been made in these EC sensors in recent years. On the contrary, the Plantower optical

particle counters have been known for their precise lower limit of detection (range of LLOD 0.08-0.24 number of particles/cm$^3$)

and lower susceptibility to the relative humidity (Bulot et al., 2020), which results in better performance of raw measurement

than in other types of OPCs (Bauerová et al., 2020; Bulot et al., 2020; Hong et al., 2021; Sayahi et al., 2019). Our results of

raw PM$_{10}$ and PM$_{2.5}$ measurements correspond with the results from other studies using the Plantower LCSs in long-term field

tests, including the slightly weaker R$^2$ values in the case of coarse particles concentrations (PM$_{10}$) than in fine particles

concentrations (PM$_{2.5}$; Bauerová et al., 2020; Hong et al., 2021; Lee et al., 2020; Sayahi et al., 2019). Overall, no significant

outliers were detected in gaseous or aerosol LCSs raw measurement during initial field comparative measurement, nor even

during the Legerova campaign or final field comparative measurement (see maximum 1-hour measured concentrations in

Tables S2-S5, Tables S15-S18 and Tables S19-S22 in the Supplement).

**4.1.2 MARS-corrected LCS measurement**

Mathematical correction of raw LCS measurements using the non-parametric MARS method achieved the best results of all

tested correction procedures (linear regression, GAM). The MARS calculation is flexible, computationally time-feasible

(calculation of the model without the interactions lasted in units of seconds, with inclusion of interactions of tens of seconds),

easy to interpret and allows taking into account various explanatory variables including their interactions (Friedman, 1991;

Keshtegar et al., 2018). In this study we used the raw LCS concentrations, TMP, RH, WV, GLRD and hour of the day as the

explanatory variables. Most of the previous studies used raw LCS measurement, TMP and RH (Considine et al., 2021; Cordero

et al., 2018; Crilley et al., 2018; deSouza et al., 2022; Jiao et al., 2016; Malings et al., 2019; Vajs et al., 2021).   Fewer studies

then also included the effect of WV, air pressure or hour of day with some mixed results (Hagler et al., 2018; Mead et al.,

2013; Munir et al., 2019; Spinelle et al., 2015, 2017). In this study the number of references to each predictor in the MARS

correction models showed that in the case of NO$_2$ the most frequently used predictors were NO$_2$ raw LCS concentrations,

TMP, WV, GLRD and the least frequently used were RH and hour of the day. In the case of O$_3$ the most frequently used were

O$_3$ raw LCS concentrations, ratio of O$_3$ and NO$_2$ LCS concentrations, TMP, RH and WV; on the other hand, GLRD and hour

of the day were, quite surprisingly, the least used predictors. In the PM measurement (both fine and coarse particles) the most

frequently used were raw LCS concentrations and then all other predictors with a similar weight. Here again, quite surprisingly

the RH was not a dominant predictor in PM correction equations. Double interactions between variables were ultimately not

included in the corrections, as they led to significant outliers in both gaseous and aerosol measurements (especially at high

peak concentrations). Nevertheless, for all measured pollutants, MARS corrections decreased MBE to close to zero in all cases

and average MAE ranged between 0.76 ppb in NO$_2$ and 3.55 µg·m$^{-3}$ in PM$_{10}$. The MARS corrections improved the

relationships with RM or EM with the average R$^2$=0.97 in NO$_2$, 0.94 in O$_3$, 0.87 in PM$_{10}$ and 0.94 in PM$_{2.5}$. The average of

generalised cross validation (GCV) error of the MARS correction models was the lowest (1.14) in NO$_2$ LCSs corrections and

the highest (27.54) in PM$_{10}$ LCSs corrections. When we compare the results of NO$_2$ and O$_3$ MARS corrections with other

statistical correction models in previous studies, we usually achieve better or similar results, e.g. maximum R$^2$=0.75 with multi-



linear regression model in Spinelle et al. (2015), $R^2=0.83$ with GAM model in Munir et al. (2019), $R^2=0.97$ with RF model in Cordero et al. (2018) and others (Barcelo-Ordinas et al., 2019). In the case of aerosol particles, Vajs et al. (2021) achieved better results ($R^2>0.90$) in the correction of $PM_{10}$ LCS measurement with different ANN or RF models and Kumar and Sahu (2021) achieved slightly better results ($R^2\geq0.98$) in $PM_{2.5}$ LCS measurement with kNN, RF, regression tree (RT) or GB methods. Conversely, Vogt et al. (2021) achieved worse results for both $PM_{10}$ and $PM_{2.5}$ in the case of correction with sensor-specific linear models (highest $R^2$ values 0.64 for $PM_{10}$ and 0.73 for $PM_{2.5}$), similarly to Kumar and Sahu (2021) with MLR correction ($R^2=0.77$) and Hong et al. (2021) with non-linear regression ($R^2=0.77$), both in $PM_{2.5}$ measurement.

When comparing the two correction procedures COR (based on initial comparison) and COR2 (based on initial and final comparison including sensor ageing), very similar results were achieved, however it is necessary to take into account the diversity of applications. While the COR method can be used to correct operationally measured LCS data, the COR2 can be applied only retroactively after the end of the entire measurement campaign.

### 4.1.3 LCS data drifts evaluation

The issue of data drifts detection has also been addressed in various studies, e.g. Malings et al. (2019) is describing the drift-adjustment based on the "Deployment Records" (DR) method, using the biases between LCSs and RM measurements during collocation (before deployment). In this method, one 'benchmark' sensor is identified from all LCSs and collocated during the whole measurement period (similarly as we collocated two LCSs in our study). Subsequent possible non-standard deviations in the LCS measurement, in particular target locations, were then assessed against the measurement bias of this benchmark LCS. This method is useful, however, as it assumes that the LCS bias is generalisable/transferable across all LCS units, which is not always the truth due to the often high individuality of sensor performances (De Vito et al., 2020; van Zoest et al., 2019). Harkat et al. (2018) described a complex framework consisting of air quality modelling, fault detection, fault isolation and reconstruction with the aim of setting the boundaries for probable and improbable LCS measurements (by using a combination of midpoint-radii PCA, generalised likelihood ratio test and exponentially weighted moving average for detecting changes in the LCSs model residuals). A simpler technique was described in van Zoest et al. (2019) where the control of LCS drifts was based on the time series of the difference/bias between the mean $NO_2$ concentrations measured by RMs placed within the area of interest and mean $NO_2$ concentrations measured by all LCSs. A zero difference was not expected here, because the LCSs were differently spaced and the difference may be subject to $NO_2$ seasonality and meteorological conditions. However, when the difference/bias began to systematically decrease or increase regardless of changes in conditions, the data drift could be indicated. As part of this study, we tried to apply similarly simple and effective data control methods, targeting the possible data drifts caused either by relocation of the LCS stations to the Prague Legerova campaign, by technical failures of the LCSs (e.g. ageing) or by loss of the MARS correction performance (the concept drift). All measurements were checked firstly by the mutual comparison of the concentration courses of LCSs located in pairs (including also the 2 collocated LCSs with RMs), secondly by the DMC method and thirdly by the final field comparative measurement at the Prague Libuš RM site. The control within pairs of LCSs or in between collocated LCS S4 and RM did not show any deviations after relocation of the LCSs to the



final deployment sites. The change in measurement performance was visually detected a few months later in the case of the

NO$_2$ S9 LCS, which drifted to gradual overestimation from September 2022 (similarly as detected in PM sensor in Sayahi et al., 2019). This was probably caused by a technical issue (different aspects discussed in Weissert et al., 2019) because the data drift was detected in both raw and corrected concentrations. This data drift was later confirmed by the DMC analysis and by final comparative field measurement (final R$^2$=0.17). The NO$_2$ S9 LCS measurement was therefore marked as invalid and was not further used for PALM model validation within the TURBAN project. Another two NO$_2$ LCSs identified based on DMC

as possibly drifted to gradual underestimation were the S11 and S12 (probably due to the loss of sensitivity of the electrochemical cell, see van Zoest et al., 2019). In these cases the data drifts were not as significant as in the S9, and during the final comparative measurement these LCSs were still performing well (R$^2$>0.81). In O$_3$ LCS measurement, a technical problem was most likely detected, as a sudden data drift (in the sense of jump to overestimation) was recorded for all LCSs from October to November 2022. Since this phenomenon also appeared in the raw measurement, the drift of the correction

concept can be ruled out (De Vito et al., 2020; Spinelle et al., 2015). During October 2022 a rapid change in air temperature (with a drop below 4 °C) occurred, which may have triggered this change of LCS measurement performance (although the correction model was trained for winter conditions during the initial test measurement; Weissert et al., 2019). In the case of PM LCS measurement no gradual or sudden data drifts were detected during the TURBAN campaign. One exception was the PM LCS S3 which was partly underestimating from the start of measurement (see corrected PM$_{10}$ and PM$_{2.5}$ concentrations in

S3 in Fig. 7). Since this LCS was leaving the initial comparison measurement at the Libuš RM earlier than the other LCSs (installed on the roof of the Karlov MS since 23 February 2022), it could be the result of an under-trained MARS correction model. However, the PM data from the S3 LCS were not marked as invalid, only as permanently underestimated, because it was not typical data drift as described above (no change of measurement performance detected during the campaign).

**4.2 Data quality of supplementary meteorological measurement**

The supplementary meteorological measurement by the MM placed in the PVK garden performed very well in comparison with the adjacent Prague Karlov MS placed on the roof of the university building throughout the entire observation campaign in expected variables (R$^2$>0.98 in TMP, RH and p). The biggest differences were in WV and WD which are typical in complex urban environments (Zou et al., 2021). The TMP profile measurement with the MTP-5-He microwave radiometer also reached very high data quality in comparison with the TMP profiles measured by radiosonde from the Prague Libuš MS. The resulting

performance was R$^2$>0.98 across different height levels and different launching times (0, 6, 12 UTC). High measurement quality (with mentioned maximum accuracy between 0.5-0.8 °C) was also described in Kadygrov et al. (2015), Kadygrov and Pick (1998) and Pietroni et al. (2014) during the comparison of the MTP-5 measurement against in situ measurements. Although for example Argentini et al. (2009) described that this type of radiometer has difficulties in detecting and measuring elevated temperature inversions, we did not observe this pattern. Other uncertainties were described by Ezau et al. (2013), who

discovered that the formation of a thin water film (of ice or, to a smaller degree, of sleet) on the surface of the older type MTP-5 sensor cover has a significant impact on the data quality of the TMP monitoring. Therefore, as part of the TURBAN





observation study, we have additionally included the comparison of MTP-5 measurements with a radiosonde in dates and times with recorded precipitations at the Prague Karlov MS. The resulting $R^2$ under rainy conditions was in our case 0.97 and higher at different height levels (with sample size n = 1172). The wind profile data measured by the Doppler LIDAR StreamLine XR

placed on the PVK roof with the VAD 6 scan mode setting were not verified within this study (against radiosonde or in situ measurement) because the different spatial conditions (and distance) in the deployment locations of the measurements in this case had a disturbing effect and such a comparison would not be representative. Nevertheless, the quality of wind profiles and root-mean-square deviations (RMSDs) from different VAD scanning programs were tested several times in comparison with radiosonde and MM, overall with the resulting $R^2>0.80$ (e.g. studies by Newsom and Krishnamurthy, 2022; Newsom et al.,

2017; Tzadok et al., 2022). Moreover, Rahlves et al. (2022) demonstrated that the VAD 6 scanning program performs more accurately in the case of WV than the VAD 24 program. Within the framework of WV profile data pre-processing, the method using the standard deviations of WV calculated in the sliding high-range window and the DMC visualisation was tested for possible flexible identification of sudden changes within the profiles. Although this method appears to be usable in most cases (with the occurrence of a few exceptions in erroneous determination), it needs to be subjected to further investigation and

testing in future.

## 4.3 Air quality and meteorological measurement within Legerova observation campaign

The results of the almost year-long observation campaign in Legerova, Sokolská and Rumunská streets and their surroundings showed that the largest load in this area is $NO_2$ pollution, due to the high daily traffic within this selected area of the Prague city centre (with the following intensity of cars per day: 37,336 in Sokolská, 35,736 in Legerova and 9,608 in Rumunská; TSK,

2023). Therefore, the daily and weekly courses of $NO_2$ concentrations corresponded well to the traffic regime in the given localities (with typical morning and late-afternoon rush hour peaks of concentrations), including the lower concentrations in background locations more distant from the emission sources (Fig. 9). The highest $NO_2$ concentrations in medians and averages behaved according to the expectations in street canyons with continuous building blocks and several traffic lights (LCSs S10 and S11 in Sokolská, S14 and S15 in Legerova and S20 and S13 in Rumunská). Locations having more open space nearby,

i.e. with a higher probability of ventilation effect, came out as moderately loaded (LCSs S12 and S18 in Sokolská school, S2 and S5 in Legerova school and S4 collocated with the Prague 2-Legerova RM). Nevertheless, the maximum 1-hour average $NO_2$ concentrations were measured by LCSs S12 and S18 placed in the Sokolská school location (Fig. 11b). Since the maximum concentration peaks were measured by both LCSs installed at different height levels, we assume that this was a reflection of some real local emission effect (i.e. a started supply car standing near the LCSs, etc.) and the random LCS error

can be ruled out. The mean and maximum $NO_2$ concentrations measured within the most loaded locations in the TURBAN observation campaign were comparable to the study of Schneider et al. (2017) focused on monitoring traffic-polluted urban sites in Oslo (FI), where the measured concentrations ranged between 42 and 63 ppb, or Moltchanov et al. (2015) in the city of Haifa (IL) with concentration peaks ranging between 50-95 ppb. On the other hand, our measurements were higher than those of Graça et al. (2023) in the city of Aveiro (PT) with $NO_2$ concentrations between 15 and 32 ppb or Wesseling et al.



(2019) measuring around 15 ppb in Amsterdam or Utrecht (NL). However, these comparisons are only indicative due to different conditions in cities.

Other interesting results within the Legerova observation campaign were reached in the case of aerosol pollution measurement. Although some daily patterns were recognisable in $PM_{10}$ and even in $PM_{2.5}$ concentrations, the concentration peaks, especially during the late afternoon, were shifted to later than the usual rush hours. The concentration peaks of $NO_2$ were observed

between 3 p.m. and 6 p.m. UTC, while the $PM_{10}$ and $PM_{2.5}$ concentration peaks usually occurred between 5 p.m. and 9 p.m. (see the detailed Fig. S49 in the Supplement). Overall there were quite low levels of PM pollution and smaller differences between different sites within the whole area of interest (medians of $PM_{10}$ ranging between 11 and 26 $\mu g \cdot m^{-3}$ and $PM_{2.5}$ between 9 and 18 $\mu g \cdot m^{-3}$). According to the measured aerosol concentrations, the most burdened locations (with medians of $PM_{10}>23$ $\mu g \cdot m^{-3}$) were LCSs S13 (in Rumunská), S11 (in Sokolská) and S14 (in Legerova; similar to $NO_2$ pollution) and the

least burdened were the background LCSs S3, S16, S19 and surprisingly even the S4 collocated with the Prague 2-Legerova RM (with medians <18 $\mu g \cdot m^{-3}$). Similarly low levels of $PM_{10}$ and $PM_{2.5}$ concentrations were measured in the city of Aveiro by Graça et al. (2023) and in Nantes (FR) by Gressent et al. (2020). One possible explanation may be the fact that nowadays, transport is no longer the main source of $PM_{10}$ and $PM_{2.5}$ particles in European cities, unlike nitrogen oxides. Transport can produce particles of a smaller size fraction ($PM_{2.5}$, $PM_1$ and smaller), which can be emitted from the incomplete combustion

of engines and emissions from brake and tire abrasion, which are part of $PM_{2.5}$ and larger size fractions. However, in both cases, the contribution of these sources forms a very small part of the total PM pollution from transport. A significant part of the pollution here is made up of coarse particles ($PM_{10}$ and larger), which settle on the road surface for a long time and are subject to resuspension (secondary dust from traffic; the amount of specific types of emissions from transport in Sokoloká and Legerova is shown in Fig. S50 in the Supplement). This also explains the similarity of $PM_{10}$ concentration trends and only

slightly higher values of concentrations measured at Legerova and at other rather background AQM stations in Prague less loaded with traffic (see Fig. S51 in the Supplement).

Similarly as in Frederickson et al. (2024) we had some difficulty in demonstrating the vertical gradient pollution effect from the LCS measurement installed at two height levels. Therefore, higher concentrations were not always measured at low heights closer to the emission sources, but sometimes even at a greater height above the ground. The vertical concentration profiles

depend mainly on atmospheric stratification, street architecture, air flow and surface properties (Frederickson et al., 2024). In connection with the atmospheric stratification, we observed high $PM_{10}$ and $PM_{2.5}$ concentrations (i.e. >40 $\mu g \cdot m^{-3}$) especially under temperature inversion conditions, even at night. From this point of view, the level of aerosol pollution was more influenced by atmospheric stratification than $NO_2$ pollution, which was more subject to the traffic regime in the streets. The highest aerosol pollution ($PM_{10}>130$ $\mu g \cdot m^{-3}$ and $PM_{2.5}>110$ $\mu g \cdot m^{-3}$) was measured temporarily in all LCS stations during the

early morning and late evening hours on 26 July 2022 according to the transported pollution from the Hřensko forest fire (confirmed from the attenuated backscatter profiles measured by the CL51 ceilometer and the Doppler LIDAR; a similar situation was detected even in other parts of the Czech Republic). Moreover, we showed a few examples of vertical stratification reconstructions and low-level jets monitored above the area of interest under temperature inversion conditions



(using the Doppler LIDAR and MWR measurement). Similar continuous TMP and wind vertical profile data above the urban
surface are not as common (Allwine et al., 2002; Kallistratova and Kouznetsov, 2012; Sánchez et al., 2022) and are very useful
in supporting advanced modelling and assessment of the impacts of air pollution and climate change in the urban environment.

## 5 Conclusion

This article describes the unique design of the TURBAN measurement campaign, including data quality control, verification
and the successful implementation of low-cost air quality monitoring network and remote sensing measurement of temperature
and wind profiles in the target location of Prague Legerova and its surroundings over 10 months (30 May 2022 – 28 March
2023). We demonstrated that if the LCSs are checked and appropriately mathematically corrected they can be used for
sufficiently reliable indicative monitoring of spatio-temporal variation of air pollution in urban environments, under high and
even low concentration conditions. The MARS correction and DMC method proved in most cases to be effective enough to
capture the known uncertainties of LCS measurements, although considering the high variability of LCS types on the market
with different instability and error probability, further testing and applications are always needed. The TMP measurement by
MWR was performed with very high data quality with a stable height range throughout the entire measurement campaign. The
quality of wind profile measurement by Doppler LIDAR was not evaluated within this study, but in comparison with the
atmosphere stratification, the data seemed reliable in most cases (under the condition of the appropriate removal of noisy data).
All the data gained during the TURBAN observation campaign are publicly available (Bauerová et al., 2024) and further used
for the new LES PALM microscale model validation. This report can serve as an important support for various modelling tools
dealing with the microscale as well as for other studies dealing with relationships among the air quality and meteorological
variables in urban environment (including the consideration of active chemicals in the air).

## Data Availability

Complete data set "TURDATA: a database of low-cost air quality and remote sensing measurements for the validation of
micro-scale models in the real Prague urban environments" is publicly available at Zenodo library at
https://doi.org/10.5281/zenodo.10655032 (Bauerová et al., 2024).

## Supplement

The Supplement contains all additional Figures and Tables, including supplementary method descriptions.



**Author contributions**

PB: Measurement campaign conceptualization & realisation, Methodology proposal, Statistical analyses, Evaluation, Manuscript draft preparation. JK: Measurement campaign conceptualization & realisation, Methodology proposal, Statistical analyses, Evaluation, Manuscript draft review & editing. AŠ: Measurement campaign conceptualization & realisation, Methodology check, Statistical analyses, Evaluation, Manuscript draft review & editing. OV: ARAMIS project leader, Supervision, Project administration, Measurement campaign conceptualization & realisation, Methodology check, Manuscript

draft review & editing. WP: Measurement campaign conceptualization & realisation, Manuscript draft review & editing. JRe: TURBAN project leader, Supervision, Administration, Funding acquisition, Measurement campaign conceptualization, Methodology check, Manuscript draft review & editing. PK: Measurement campaign conceptualization, Validation. JG: Measurement campaign conceptualization, Validation. HŘ: Measurement campaign conceptualization, Validation, Manuscript draft review & editing. MBu: Measurement campaign conceptualization, Validation. KE: Statistical analysis control. MBe:

Measurement campaign conceptualization, Manuscript draft review & editing. JRa: Measurement campaign conceptualization. VF: Measurement campaign conceptualization, Methodology control. RJ: Measurement campaign conceptualization, Emission data preparation, TURBAN project website preparation. IE: Measurement campaign conceptualization.

**Competing interests**

The authors declare that they have no known competing financial interests or personal relationships that could have appeared
to influence the work reported in this paper.

**Disclaimer**

Publisher's note: Copernicus Publications remains neutral with regard to jurisdictional claims made in the text, published maps, institutional affiliations, or any other geographical representation in this paper. While Copernicus Publications makes every effort to include appropriate place names, the final responsibility lies with the authors.

**Acknowledgments and financial support**

The measurement campaign and data processing was financed by the Norway Grants and Technology Agency of the Czech Republic (TA CR) project TO01000219 "TURBAN": Turbulent-resolving urban modelling of air quality and thermal comfort; methods used for the data processing were developed within the Technology Agency of the Czech Republic (TA CR) project SS02030031 "ARAMIS": Air quality Research, Assessment and Monitoring Integrated System. Furthermore, we are grateful

to Jan Šilhavý, Zdeněk Běťák and Luboš Vrána for technical support, to the Office of the Municipal District of Prague 2, to the Prague Waterworks and Sewerage Company, to the Czech Chamber of Authorised Engineers and Technicians active in



construction, to Le Palais Art Hotel Prague and to the elementary school and language school VĚDA for cooperation in the placement of the measuring devices. We are also grateful to Erin Naillon for proofreading the manuscript.

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
