# Peer review of "Measurement report: A complex street-level air quality observation"

_EGUsphere, 2024_

## Referee Comment (RC1)

1. General comments

The paper "Measurement report: TURBAN observation campaign combining street-level low-cost air quality sensors and meteorological profile measurements in Prague" outlines very well the strategy used to verify and quality control the data of low-cost sensors networks in different conditions against consecrated and more robust techniques.

The manuscript is overall well written and addresses relevant issues.

2. Specific comments

This paper is a response to an actual necessity of data availability at a higher spatial resolution in cities.

A further improvement is necessary, to became more scientific appropriate (e.g "similar course of concentrations over time", "In this context, a very appropriate question is offered, namely what is sufficiently long field comparative measurement? The answer to this question is not clearly defined anywhere. Overall the recommendation based on the experience of different studies is, the longer the better.")

Please avoid abbreviations on Schemes or graphs (e.g Figure 5) and improve the contrast and quality of images.

The paper is rather long and difficult to follow, due to multiple details. I think the authors should reconsider the paper organization and the beneficial of pollution events and remote sensing data in this manuscript.
A stronger conclusion, more focused on the low sensors networks necessary characteristic, testing parameters, regular checks and drifts supervision should be added.

3. Technical corrections
   - Please avoid abbreviations on Abstract and graphical abstract
   - It is worthy to consider also periodical checks for LCS stability overtime after the laboratory or field calibration (line 65)
   - Please include information related to the calibration and field calibration concentrations interval and subsequent consequences
   - Line 164: mention the heights interval, as included on the Table 1
   - Add details related to the linear response of the sensors on diverse concentration intervals
   - Add more comments related to the relationship of concentrations measured by LCS and by reference monitor, e. g slope difference between raw and corrected data
   - Add more insights related to:
     - Figure 6 A: high difference between RM and S11 concentrations for example, more than 50%
     - Figure 6 C&D: 19-22.01.2022 high difference between EM and LCS concentrations, different variability
   - Figure 7 caption, add explanation for grey dots
   - Figure 7 and 8: include the negative values as well where is the case to explain the median values.

- Line 359: explain the large variability of LCS pairs heights, as described in Table 1 and related uncertainties
- Section 2.3.2 the comparison of MWR to radiosonde temperature profiles were performed for multiple locations, please mention here at least this and discuss the findings in relation with other studies at line 690
- If possible please discuss the weekly variation of pollutants in relation with other studies in Prague region
- A short comment on diurnal variation will be also valuable, some sensors concentrations are not showing the typical diurnal variation, maybe a correlation with the location
- Line 500 and Figure S37, it can be seen the low SNR on ceilometer attenuated backscatter due to low clouds, comment related to this finding, moreover please add y axes label
- Line 550 the pollution events seems to be over a very short time frame (2-4 hour), please explain the long-range aspect in this case
- Line 730 please include an explanation for differences in $NO_2$ and PM diurnal variations and peaks hour
- Please mark with different colour or marker the sites locations (e.g background, traffic) in all graphs to be easier to follow
- Line 750 the concentrations at different heights should be considered using also the measurements uncertainties.

---

## Author Comment (AC1)

*First of all, we would like to thank the reviewer for reading the manuscript and for the valuable comments and suggestions for article improvements. The manuscript has been extensively revised. Due to the significant changes in the article, we have decided to modify the title of the article to better reflect its focus. New title "Measurement report: A complex street-level air quality observation campaign in the heavy traffic area utilizing the multivariate adaptive regression splines method for field calibration of low-cost sensors".*

*We believe that these changes will improve the readability of the article.*

*Listed below are our answers and the changes made to the manuscript according to the remarks and suggestions given by the reviewer. The comment of the reviewer is listed below, our responses are highlighted in blue italics.*

**Reviewer #1**

1. General comments

The paper "Measurement report: TURBAN observation campaign combining street-level low-cost air quality sensors and meteorological profile measurements in Prague" outlines very well the strategy used to verify and quality control the data of low-cost sensors networks in different conditions against consecrated and more robust techniques. The manuscript is overall well written and addresses relevant issues.

2. Specific comments

This paper is a response to an actual necessity of data availability at a higher spatial resolution in cities. A further improvement is necessary, to became more scientific appropriate (e.g. "similar course of concentrations over time", "In this context, a very appropriate question is offered, namely what is sufficiently long field comparative measurement? The answer to this question is not clearly defined anywhere. Overall the recommendation based on the experience of different studies is, the longer the better.")

*The aforementioned parts of the text have been rephrased and edited to contain only clear facts. See lns 72-75 on p3 and lns 301-311 on p12.*

Please avoid abbreviations on Schemes or graphs (e.g. Figure 5) and improve the contrast and quality of images.

*Thank you for your notice, the abbreviations have been explained in the graphical abstract (see p2) and in Figure 4 on p11. The contrast and quality was improved in following figures: Fig. 7 on p17, Fig.8 on p18 and Fig.11 on p21.*

The paper is rather long and difficult to follow, due to multiple details. I think the authors should reconsider the paper organization and the beneficial of pollution events and remote sensing data in this manuscript.

*Yes, we acknowledge that the original version of the manuscript was too long and not well structured. The manuscript has been extensively revised (see document with marked changes).*

*The revised version of the article focuses mainly on the methods of sensor data quality verification, correction and continuous control, as well as the results of the Legerova measurement campaign and interesting episodes related to air quality. As recommended, we have moved the description of the methodology, results and discussion regarding the accompanying meteorological measurements to the Supplement. On the other hand, we have moved some important information concerning sensor measurements to the text of the article. We strongly believe that the changes made have improved the readability of the article.*

A stronger conclusion, more focused on the low-cost sensor networks necessary characteristic, testing parameters, regular checks and drifts supervision should be added.

*The conclusion section was rewritten according to the reviewer's recommendations and changes made in the manuscript during revision. See lns 667-693 on p27.*

3. Technical corrections

- Please avoid abbreviations in Abstract and graphical abstract

  *Both modified according to the reviewer's note.*

- It is worthy to consider also periodical checks for LCS stability overtime after the laboratory or field calibration (line 65)
  *You are correct. Thank you for that comment. We have added the following sentence to the introduction "...; 3) periodically check the sensors' performance over time, if possible repeat comparative measurement at the reference station (identification of data drifts)." (see lns x-y in revised version.*

- Please include information related to the calibration and field calibration concentrations interval and subsequent consequences

  *In view of this remark, we have added to the text of the introduction the indisputable advantage of performing laboratory calibration ("The advantage of laboratory calibration is the possibility to identify possible differences in sensor response to different concentration levels." see lns 64-66 on p3).*

  *As we have no experience with calibrating sensors in the laboratory, we cannot make a relevant assessment of the uncertainties arising from undergoing/not undergoing this process. This topic is the subject of other articles.*

- Line 164: mention the heights interval, as included on the Table 1

  *The height intervals were added to the text, see lns 153-160 on p5.*

- Add details related to the linear response of the sensors on diverse concentration intervals
  *As we did not calibrate the sensor units in the laboratory as part of this study, we are unfortunately unable to determine the linear response to different concentration intervals. Based on the field comparative measurements, we were only able to determine the linear response over the entire measurement range. The results of the linear*

*regression between the reference and sensor measurements ($R^2$, slope, intercept and MBE) have been added to the text in the results section, see lns 301-322 on p12.*

- Add more comments related to the relationship of concentrations measured by LCS and by reference monitor, e. g slope difference between raw and corrected data
*The correlation coefficients and standard deviations showing the difference between raw and corrected sensor concentrations and reference measurements are newly shown in Fig. 5 in the revised version of the manuscript (see p14). Results of the linear regression between the reference and sensor measurements ($R^2$, slope, intercept and MBE) have been added to the main text in the results section and are further discussed in the discussion, see lns 301-322 on p12 and p23-24.*

- Add more insights related to:
  - Figure 6 A: high difference between RM and S11 concentrations for example, more than 50%
    *Yes, sensor S11 was one of those with larger deviations in the raw $NO_2$ measurement (along with other sensors S3 and S14). Including the fact that the S11, S3 and S4 sensors showed slightly different performance even after applying the MARS correction. This fact is also mentioned in the revised version of the manuscript in the results section (see lns 318-319 on p12) and shown in Taylor diagrams with standard deviations and correlation coefficients of individual sensors before and after correction (see Fig. 5 on p14). The original Fig. 6 was moved to the supplement (newly as Fig. S7 in the revised version of the Supplement).*

  - Figure 6 C&D: 19-22.01.2022 high difference between EM and LCS concentrations, different variability
    *Again, you are right. The high variability of measurement deviations is quite common in low-cost sensors. The worse measurement performance in coarse particles $PM_{10}$ (shown former Fig. 6c) than in fine particles $PM_{2.5}$ (shown former Fig. 6d) is described in many previous studies, see part of the introduction "The mass concentration of the coarse fraction of aerosol particles ($PM_{10}$) is usually burdened by weaker measurement performance and by the greater probability of measurement error with respect to relative humidity than the fine fraction $PM_{2.5}$ (Bauerová et al., 2020; Crilley et al., 2018; Tagle et al., 2020; Tryner et al., 2020)" (lns 54-57 on p3). This was also the case in PM measurement in our study, as described in the results and discussion section.*

  - Figure 7 caption, add explanation for grey dots
    *An explanation of black dots shown in boxplots was added to the figure captions (see Figure 6 on p15 in the revised version of the manuscript and Figure S28 and S42, on p27 and p41, respectively in the revised Supplement).*

  - Figure 7 and 8: include the negative values as well where is the case to explain the median values.
    *Information about the occurrence of weakly negative concentration values in the dataset after MARS correction is mentioned in the results section lns 321-323 on p12: "However, after correction, some initially very low concentrations turned into weakly negative values: for gaseous pollutants, they constituted less than 0.3 % and*

*for aerosol less than 2.6 % of the whole testing dataset (part of the summary statistics in Tables S12–S15 in the Supplement)".*

*With regard to the reviewer's comment this information has been added also to the captions of Figure 6 in the revised version of the manuscript (p15) and Figure S28 and S42, on p27 and p41, respectively in the revised Supplement).*

*The fact that the boxplots show medians was already stated in the original descriptions of boxplot figures. The medians were used because they are not affected by outliers in the dataset (as this is a well-known fact, we took the liberty of not explaining this further).*

- Line 359: explain the large variability of LCS pairs heights, as described in Table 1 and related uncertainties
*The variation in sensor placement heights (installed in pairs or individually) was determined by the installation options at each site. As the main objective of the campaign was to obtain the most reliable data with high spatial and temporal resolution within the area of interest, we do not consider this to be a problem. The data are to be used for the purpose of validating microscale models; from this point of view, variability in the heights of the measurement points is instead appropriate (model outputs can be validated at specifically defined heights). We are not able to determine the specific uncertainties in sensor measurements at different heights within the scope of this study.*

- Section 2.3.2 the comparison of MWR to radiosonde temperature profiles were performed for multiple locations, please mention here at least this and discuss the findings in relation with other studies at line 690
*In the case of temperature profiles, data from two different locations during the Legerova campaign were compared, i.e. data from the MWR located at the Prague Karlov meteorological station with data from radiosondes launched at the Prague Libuš MS. In view of your comment, we have modified the description of methods used as follows: "The vertical profiles of TMP measured by the MWR in the Prague Karlov MS were indicatively checked against TMP vertical profiles measured by radiosonde launched from the distant Prague Libuš MS during the period from 25 February 2022 to 24 March 2023…" (see lns 198-200 on p8 in the revised version of the Supplement). This fact was also mentioned in the discussion section: "Although it should be noted that our data verification was not within the co-location of the instruments, but they were about 8 km apart by air." see lns 712-713 on p49 in the revised Supplement.*

- If possible please discuss the weekly variation of pollutants in relation with other studies in Prague region
*We are currently not able to do more comparisons than those already commented in the manuscript (see comparison of measurements at different traffic reference monitoring stations in Table S1 on p1 and in Fig. S58 on p50 in the Supplement). Although we have experience with measurements (and model validation) in another part of Prague (see Resler et al., 2021), the results are not comparable due to the different setup of the measurement campaign (only short-term measurements and with much less spatial variability). The design of the Legerova measurement campaign was specifically designed with these experiences from the previous campaign in mind (need for meteorological and air quality high spatiotemporal data).*

- A short comment on diurnal variation will be also valuable, some sensors concentrations are not showing the typical diurnal variation, maybe a correlation with the location

  *Diurnal variations of concentration of monitored pollutants are described in the result section 3.2 (p15-16) and are further discussed in section 4.2 in the manuscript (p25-27). Sensors with different (weaker) diurnal variations were placed at the background locations. Some small differences can be observed between the $NO_2$ concentration trends within different streets (probably in response to a slightly different traffic load). However, we are unable to confirm this within the design of this study. This is more of a task for subsequent microscale model calculations, which may involve much more input data (such as building heights, surface properties, heat flow, traffic load, etc...).*

- Line 500 and Figure S37, it can be seen the low SNR on ceilometer attenuated backscatter due to low clouds, comment related to this finding, moreover please add y axes label

  *Thank you for this remark. We have added the information about the occurrence of low clouds over the aerosol layer in the morning hours to the caption of both figures, i.e. Fig. S43 and S44 in the revised version of the Supplement, both on p42. The y axis label has been added to the Fig. S43.*

- Line 550 the pollution events seems to be over a very short time frame (2-4 hour), please explain the long-range aspect in this case

  *You are right, the pollution events shown in the examples were in a short time frame. In our experience, these dynamics are not that rare in the atmosphere. In particular, during the rapid morning boundary layer transition and ground-level inversion decay, polluted air from the residual layer reaches the ground, which can cause relatively rapid concentration changes (in the form of short-term spikes). Whether the pollution in the residual layer is the result of an accumulation from the previous day or from long-distance transport we are not able to determine in these cases (except for notable episodes such as the forest fire in Hřensko).*

- Line 730 please include an explanation for differences in NO2 and PM diurnal variations and peaks hour

  *As mentioned in a previous response, we do not have enough information to explain the reasons for these differences in concentration peaks within the observed area. We only provide results on measured air quality with measured meteorological data. The reasons for the dynamics of pollution at individual sites can only be relevantly assessed within the framework of comprehensive model calculations on a wide spectrum of input data, which we do not have in the context of this study.*

- Please mark with different colour or marker the sites locations (e.g background, traffic) in all graphs to be easier to follow

  *Thank you for this suggestion. We have marked the background sites (and sensors) with asterisk in figures and tables where appropriate, i.e. Fig. 1 (p7), Fig. 7 (p17), Fig. 8 (p18), Fig. 9 (p19) and in Tables 4 (p17) and 5 (p18) in the revised version of the*

*manuscript; as well as in Fig. S40 and S41 (p39-40), Tables S20 and S21 (p40-41) in the revised version of the Supplement.*

- Line 750 the concentrations at different heights should be considered using also the measurements uncertainties

  *As already mentioned before, we are not able to determine the specific uncertainties in sensor measurements at different heights within the scope of this study. Answering this question would be a possible topic for some future studies - assuming a different design, where the sensors are verified against a reference measurement located at different heights a.g.l.*

*References:*

*Resler, J., Eben, K., Geletič, J., Krč, P., Rosecký, M., Sühring, M., Belda, M., Fuka, V., Halenka, T., Huszár, P., Karlický, J., Benešová, N., Ďoubalová, J., Honzáková, K., Keder, J., Nápravníková, Š., and Vlček, O.: Validation of the PALM model system 6.0 in a real urban environment: a case study in Dejvice, Prague, the Czech Republic, Geoscientific Model Development, 14, 4797–4842, https://doi.org/10.5194/gmd-14-4797-2021, 2021.*

---

## Author Comment (AC2)

*First of all, we would like to thank the reviewer for reading the manuscript and for the valuable comments and suggestions for article improvements. The manuscript has been extensively revised. Due to the significant changes in the article, we have decided to modify the title of the article to better reflect its focus. New title "Measurement report: A complex street-level air quality observation campaign in the heavy traffic area utilizing the multivariate adaptive regression splines method for field calibration of low-cost sensors".*

*We believe that these changes will improve the readability of the article.*

*Listed below are our answers and the changes made to the manuscript according to the remarks and suggestions given by the reviewer. The comment of the reviewer is listed below, our responses are highlighted in blue italics.*

**Reviewer 2:**

The topic of the manuscript is both interesting and relevant to ACP readers. However, significant revisions to the presentation of the methods and results are necessary before the validity of the findings can be assessed and their relevance fully evaluated.

*We would like to thank the reviewer 2 for his valuable remarks and recommendations. The manuscript has undergone extensive revisions, including incorporation of most of the comments. We improved the description of the used methods and we moved some important results related to measurements with low-cost sensors to the article. On the contrary, most of the information related to the supplementary meteorological measurement has been moved to the Supplement. We strongly believe that the changes made have improved the readability of the article.*

Abstract is somewhat confusing, more like a list of unrelated sentences and not a concise overview. Please rewrite.

*The abstract has been completely rewritten. Please see the revised version of the manuscript.*

Basics of MARS method (why, how, the elements of the model, meaning of splines) should be described as it cannot be considered as commonly known method.

*The description of the MARS method has been modified in the methodology section in the revised version of the manuscript (section 2.3, lns 243-247 on p10), especially by adding the summary Table 3 (p11) with the design of the COR and COR2 MARS models. The detailed description of the MARS method principles has been added to the revised version of the Supplement to the section S2.3.1 (p3-4). The aspects why we used this method are mentioned in the introduction and in discussion (see lns 92-101 on p3-4 and lns 518-528 on p23) in the revised version of the manuscript.*

The method description lacks sufficient detail to allow for replication of the analysis, which should be the standard level of detail. Conversely, the methods section includes numerical information that would be better suited to the results section. This misplacement makes the methods section harder to read. When transferring these details to the results section, explain the significance of the numbers (e.g., CVs, R²) and the narrative they convey.

*The methods section has been revised as recommended. The structure of the chapter has been changed for better clarity with a focus on methods used for the LCS network (methods related to meteorological measurements have been moved to the Supplement). In addition, the numerical information has been moved to the results section 3.1 in the revised manuscript.*

*An explanation of the contribution of the metrics used has been provided in the methodology section 2.3, lns 228-242 on p9: "Summary statistics included the coefficient of variation (CV) to express mean precision of LCS measurements during field measurements, along with mean, median, standard deviation (SD), and parameters derived from regression analyses: intercept (a), slope (b), coefficient of determination (R²), Williamson-York regression parameters (a, b, using the maximum given RM, EM and LCS uncertainties; according to Cantrell, 2008), mean bias error (MBE) and root mean square error (RMSE)".*

*Statistical significance has been added to the correlation coefficients mentioned in the results in lns 306-307 on p12 (as all relationships presented were statistically significant at the $p<0.05$ level, they are not reported further).*

It should be discussed whether linearity can be assumed in the data and whether the linear regression model is therefore appropriate. If linearity can be assumed I would recommend considering more robust fitting methods, as the ordinary least squares has been shown to be ill-suited for many types of atmospheric data (see e.g. Cantrell, 2008 and Mikkonen et al. 2019).

*We compare the reference measurement method with the values measured by the sensor without correction (raw data) and the values corrected using the MARS method (corrected data). In accordance with the commonly used methodological approach, we perform a linear best fit of the scatter plot of the reference method versus the sensor measurement. However, we thank you for your valuable comment at this point, because we acknowledge that both methods (sensor and reference monitors) are subject to error. Therefore we have additionally calculated the Williamson-York Iterative Bivariate regression (see lns 236-238 on p9 in Sect. 2.3 in the revised version of the manuscript) and showed these results as the part of LCS summary statistics related to the initial and final field comparative measurement (see Tables S12-S15 and Tables S16-S19 in the revised version of the Supplement). The bivariate regression was calculated according to the recommended publication Cantrel (2008). The used values of variance were calculated from the maximum possible uncertainties specified for the measurement method (15 % for $NO_2$ and $O_3$ RM measurement, 25 % for $PM_{10}$ and $PM_{2.5}$ EM measurement and 30 % for LCS measurements) since we do not have the exact (laboratory-verified) values of uncertainties, especially in case of LCSs.*

Section 2.3.2 could also be improved by adding subtitles or breaking it into smaller, thematically organized segments for better readability.

*We agree with this comment. Section 2 (Materials and methods) has undergone extensive changes and shortening. Newly, there are only three main subsections: 2.1 Study area and experimental design, 2.2 Technical specification of instruments used and measurement*

*methods, 2.3 Data processing and statistical analyses. We believe that changes made improved the readability of this section.*

Describe clearly in the main text what are the COR and COR2 methods as your results are strongly dependent on them. Show an example of a correction model in the main text and justify the form of the model. The meaning of the equations in Table S14 is not fully clear even for experienced statistician, let alone the average reader of the ACP.

*Thank you for this valuable remark. The description of COR and COR2 method was modified in the text of method section 2.3 and improved by adding Table 3 with the design of the COR and COR2 MARS models. We believe that changes made will help readers to better understand the summary statistics of correction model outputs shown in Tables S2-S10 in the Supplement (i.e. Tables SS6-S14 in the original version of the Supplement), which are important in our opinion for demonstrating the performance of the individual correction models.*

The manuscript leans too heavily on the massive Supplement. Reconsider which result tables and figures you are showing in the main text. If important results are shown only in 51-page supplement, it will not be found by readers.

*You are correct, thank you for this important note. As already mentioned before, we moved some important results related to low-cost sensor measurements to the main text of the article, including added Figure 5 (14), which represents the quality of the sensor measurement before and after correction.*

Minor remarks

There are two sections numbered as 2.3.2

*We apologize for this typo in the original manuscript. This error is no longer repeated in the revised version.*

Section 4.1.2: Comparing R2 values from different studies is not meaningful as the value can be calculated in numerous ways which are not necessary comparable.

*Thank you for this remark. Coefficient of determination ($R^2$ resulting from linear regression between reference and sensor measurement) is one of the most common metrics used to express sensor measurement performance (before and after various corrections applied) in numerous studies (e.g. Vajs et al. 2021, Kumar and Sahu, 2021 or Borrego et al., 2016 and others). Therefore, we decided to leave $R^2$ based comparison with other studies in the discussion section. However, as part of the manuscript revision, we verified that the studies we compare with in the discussion used really $R^2$ from linear regression between the reference and sensor measurements.*

As different calibration methods are discussed in the manuscript, it would also be interesting to hear what the authors think on use of dynamic models, like in Zaidan et al. (2020), which are able to account for evident autoregression in the data.

*We thank the reviewer for recommending this article. The authors achieved interesting results using very comprehensive autoregressive models. Although we understand the direction that the paper suggests, for the purposes of our study, the choice to correct the LCS network using the MARS method was a fast yet effective method to validate the measured data collected for the purpose of microscale model validation. However, we are impressed by this method and can focus on testing it in future campaigns.*

Lines 737-738: stating that transport is not the main source of PM10 and PM2.5 in European cities needs reference.

*This statement has been edited with adding the reference: "These results may suggest that with the current development of cars in recent years, transport might not be the main source of aerosol pollution in European cities, unlike nitrogen oxides (see for example Scerri et al., 2023)" (lns 641-643 on p26).*

*Scerri, M. M., Weinbruch, S., Delmaire, G., Mercieca, N., Nolle, M., Prati, P., and Massabò, D.: Exhaust and non-exhaust contributions from road transport to PM10 at a Southern European traffic site, Environmental Pollution, 316, 120569, https://doi.org/10.1016/j.envpol.2022.120569, 2023.*

References

Cantrell, C. A.: Technical Note: Review of methods for linear least-squares fitting of data and application to atmospheric chemistry problems, Atmos. Chem. Phys., 8, 5477–5487, https://doi.org/10.5194/acp-8-5477-2008, 2008.

Mikkonen, S., Pitkänen, M. R. A., Nieminen, T., Lipponen, A., Isokääntä, S., Arola, A., and Lehtinen, K. E. J.: Technical note: Effects of uncertainties and number of data points on line fitting – a case study on new particle formation, Atmos. Chem. Phys., 19, 12531–12543, https://doi.org/10.5194/acp-19-12531-2019, 2019.

Zaidan, M. A. et al., "Intelligent Calibration and Virtual Sensing for Integrated Low-Cost Air Quality Sensors," in IEEE Sensors Journal, vol. 20, no. 22, pp. 13638-13652, 15 Nov.15, 2020, doi: 10.1109/JSEN.2020.3010316.